# Random Process Flow Matching:
# Generative Implicit Representations of Multivariate Random Fields

**Julien Lalanne** [1 2 3]   **David Picard** [3]   **Lionel Boillot** [2]   **Lina-María Guayacán-Carrillo** [1]   **Leon Barens** [2]
**Jean-Michel Pereira** [1]

## Abstract

Generative modeling provides a powerful framework for learning data distributions. These models initially relied on probabilistic methods such as Gaussian Processes (GP) for uncertainty-aware predictions and shifted towards larger trainable models to learn more complex distributions. In this work, we introduce *Random Process (RP) Flow*, a Flow Matching-based framework that represents the vector field as a neural implicit function. Unlike modern generative methods, our setting involves a single observed field, from which only sparse measurements are available. RP Flow uses Random Fourier Features to learn an implicit signal representation that can be queried at any arbitrary location from a limited set of observations, while encoding uncertainty through ensemble sampling. We propose constructing a Bayesian posterior by GP regression in the source space to generate high-quality samples. Our empirical results demonstrate that this framework generates realistic samples along with calibrated uncertainty estimates, even under challenging conditions such as high frequency, high sparsity, or high dimensionality. These findings position RP Flow as a milestone towards generative models for reconstruction tasks where data is scarce and uncertainty must remain traceable.

## 1. Introduction

Generative models have significantly improved our ability to learn complex data distributions. In particular, diffusion ([Sohl-Dickstein et al., 2015](#); [Ho et al., 2020](#); [Song et al.,](#)

[1]Navier, CNRS, Univ Gustave Eiffel, ENPC, IP Paris, France [2]TotalEnergies, OneTech, France [3]LIGM, CNRS, Univ Gustave Eiffel, ENPC, IP Paris, France. Correspondence to: Julien Lalanne <julien.lalanne@enpc.fr>.

*Proceedings of the 43$^{rd}$ International Conference on Machine Learning*, Seoul, South Korea. PMLR 306, 2026. Copyright 2026 by the author(s).

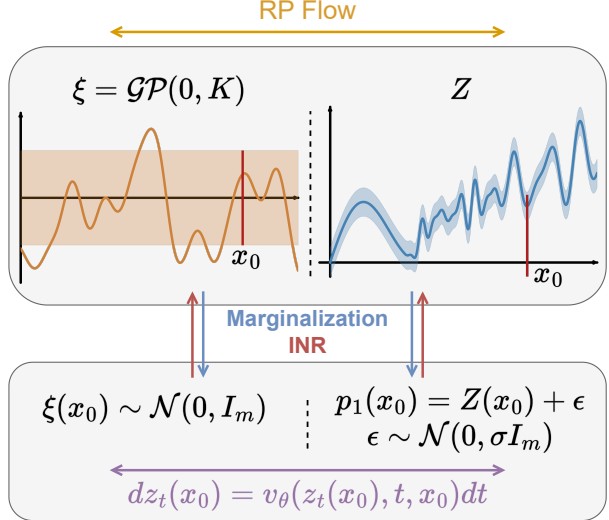

*Figure 1. Framework of RP Flow.* The transport between the random processes is implicitly learned by marginalization of the source and target processes at each training position, using a position-conditioned Flow Matching model. Aleatoric uncertainty is modeled with random noise $\epsilon$ in the target process. Posterior processes are later defined using Gaussian Processes in the source space to enable conditioning on observations.

2021a) and score-based models ([Song et al., 2021b](#)) have demonstrated remarkable success in high-dimensional domains such as image synthesis ([Dhariwal & Nichol, 2021](#); [Rombach et al., 2022](#); [Lugmayr et al., 2022](#)), video generation ([Ho et al., 2022](#); [Blattmann et al., 2023](#)) or audio modeling ([Chen et al., 2021a](#); [Kong et al., 2021](#)) by learning an iterative mapping from data to a simple isotropic Gaussian distribution. Due to their simplicity and robustness ([Chung et al., 2023](#)), these methods gained significant popularity for many real-world industrial application, including medical imaging ([Moghadam et al., 2023](#); [Lyu & Wang, 2022](#)), weather forecasting ([Price et al., 2023](#); [Andrae et al., 2025](#)) and geosciences ([Durall et al., 2023](#); [Federico & Durlofsky, 2025](#)). Conditional Flow Matching ([Lipman et al., 2023](#); [Ma et al., 2024](#)) has more recently been introduced as a generalized framework of diffusion by constructing a probability path between a known source distribution and a target data distribution. This approach enables straightforward likelihood estimation, invertible mappings and generalized

transformations between arbitrary distributions.

While powerful, these models usually treat entire signal observations as independent samples drawn from an underlying distribution. Although this formulation allows the models to efficiently learn the complex structure of the data, it also requires access to large datasets, which is not feasible for many real-world scenarios where data is sparse, incomplete or irregularly sampled.

In this work we propose Random Process (RP) Flow, a novel framework that models signals as realizations of random fields. This approach allows to model the marginal distribution of a single signal at observed locations using Flow Matching, while simultaneously capturing its structure through an implicit neural representation. By expressing the learning task as a transductive self-supervised problem (Gui et al., 2024; Tachella & Davies, 2026), we eliminate the need for large-scale training datasets, enabling interpolation and uncertainty quantification problems solvable in domains where comprehensive data acquisition is impractical or impossible. Our method creates a bi-directional mapping between a predefined Gaussian source space and the target signal space, that can be conditioned on collocation points. By leveraging Gaussian Process Regression in conjunction with the invertibility of the Flow Matching ODE, our method defines a posterior source process that can be used to condition on observations when evaluating RP Flow at new locations, resulting in high-quality samples from the posterior target process. We first validate our method on transductive image regression tasks, where it demonstrates strong reconstruction accuracy and well-calibrated uncertainty estimates. We then evaluate its performance on a seismic data interpolation task, a challenging three-dimensional problem that can be formulated as a two-dimensional multivariate random field. In this high-dimensional and structured setting, RP Flow achieves state-of-the-art performance.

Our main contributions are summarized as follows:

- We introduce RP Flow, a novel formulation of Flow Matching for spatial processes, using Implicit Neural Representations to model spatially-coupled probability paths. This approach enables learning from a single realization observed on sparse locations.

- We demonstrate that the learned transport map preserves both the regularity and the statistics of the source process.

- We propose a posterior-sampling method based on Gaussian process regression in the source space, enabling conditioning on observed data while ensuring unbiased predictions.

- We empirically show that RP Flow produces high-quality interpolations and provides calibrated uncertainty estimates, even in high-frequency and high-dimensional multivariate settings.

## 2. Background and Related Work

Our approach builds upon recent advances in generative modeling, probabilistic inference, and neural representations. In this section, we outline the key concepts used in our framework: Conditional Flow Matching (CFM), Gaussian Process Regression (GPR), and Implicit Neural Representations (INRs).

**Conditional Flow Matching (CFM)** is a simulation-free framework for generative modeling that trains Continuous Normalizing Flows (CNF) (Chen et al., 2021b) by regressing vector fields that transport a source distribution to a target distribution along a fully-described probability path (Lipman et al., 2023). Let $p_0$ be a source distribution (e.g., $\mathcal{N}(0, I)$), and let $p_1$ be the target data distribution. CFM assumes a continuous probability path $(p_t)_{t \in [0,1]}$ interpolating between $p_0$ and $p_1$, governed by a velocity field $v_t$ via the continuity equation:

$$\partial_t p_t(z) + \nabla \cdot v_t(z_t) p_t(z_t) = 0 \qquad (1)$$

CFM learns a parameterized velocity field $v_\theta(z, t)$ such that the trajectories $z_t$ of particles initialized from $z_0 \sim p_0$ evolve according to the ordinary differential equation

$$dz_t = v_\theta(z_t, t)dt, \quad z_0 \sim p_0 \qquad (2)$$

ensuring that the evolving distribution of $z_t$ matches the path $(p_t)$ from $p_0$ to $p_1$.

During training, we consider a family of conditional distributions $p(z|c)$, where $c$ denotes a conditioning variable. The conditional probability path $p(z|t, c)$ is then governed by a conditional velocity field $v_t^{\text{cond}}(z, c)$. CFM learns $v_\theta(z, t)$ by minimizing the expected squared error between the model and the true conditional velocity field:

$$\mathcal{L}_{\text{CFM}}(\theta) = \mathop{\mathbb{E}}_{\substack{t \sim \mathcal{U}[0,1], \\ z \sim p(z|t,c)}} \left\| v_\theta(z, t) - v_t^{\text{cond}}(z, c) \right\|^2 \qquad (3)$$

This regression-based objective avoids the need to evaluate densities or compute gradients of log-likelihoods, making it particularly suitable for high-dimensional or structured data. Moreover, CFM supports fast sampling by solving the ODE (2) forward in time, as well as reverse sampling by solving it backward in time.

This framework positions CFM as a powerful alternative to score-based diffusion models (Ho et al., 2020; Song et al., 2021b), particularly in scenarios where reverse sampling must remain doable or where the source distribution deviates from a standard Gaussian. Recent developments have extended CFM to incorporate optimal transport paths (Tong et al., 2024; Pooladian et al., 2023; Kornilov et al., 2024),

resulting in more efficient and stable training dynamics. Other approaches have investigated the use of alternative source distributions (Kollovieh et al., 2025) or leveraged the learned map as a prior for solving inverse problems (Zhang et al., 2024). However, samples from the data distribution are typically assumed to be independent, and using CFM for interpolation usually requires access to a large dataset of reconstructed samples. We aim to employ the CFM framework in a self-supervised manner using a single incomplete observation, removing the need for additional training data.

**Gaussian Process Regression (GPR)** is a class of nonparametric Bayesian models, particularly well-suited for spatial data due to their ability to encode prior beliefs about smoothness and correlation through a kernel function. A GP defines a distribution over functions such that any finite collection of function values follows a multivariate Gaussian distribution. The choice of kernel $K$ determines the properties of the sample paths, including regularity, stationarity and isotropy (Rasmussen & Williams, 2006).

In GPR, given a set of observations $\mathcal{D} = \{(x_i, z_i)\}_{i=1}^N$ where $z_i \sim \mathcal{N}(Z(x_i), \sigma^2 I_m)$, the goal is to infer a posterior distribution over functions conditioned on the observations. Assuming a prior $Z \sim \mathcal{GP}(0, K)$, the posterior process $Z(x)|\mathcal{D}$ remains a Gaussian Process with mean and covariance functions given by:

$$
\begin{aligned}
\mu(x) &= K(x, X)[K(X, X) + \sigma^2 I_m]^{-1} z \\
\Sigma(x, x') &= K(x, x') \\
&\quad - K(x, X)[K(X, X) + \sigma^2 I_m]^{-1} K(X, x')
\end{aligned}
\tag{4}
$$

This posterior provides both predictions and uncertainty estimates at arbitrary test locations, making GPR a powerful tool for interpolation and probabilistic modeling (Capone et al., 2023).

However, despite their interpretability, GPR faces scalability challenges in high-dimensional settings due to the cubic cost of matrix inversion and the need to store dense covariance matrices. Moreover, for many applications, the data itself cannot be considered Gaussian ; and standard kernels such as the squared exponential assume stationarity and isotropy, which may not hold in complex spatial domains.

**Implicit Neural Representations (INRs)** are models that encode signals such as images, shapes, or fields as continuous functions parameterized by neural networks. Instead of storing discrete samples on a grid, INRs learn a mapping from spatial coordinates to signal values. This paradigm enables high-resolution reconstruction, memory efficiency (Tancik et al., 2021; Lee et al., 2021), and spatial continuity, making INRs particularly suitable for modeling spatial processes and physical fields (Tancik et al., 2020;

Sitzmann et al., 2020). Formally, an INR defines a function $f_\theta : \mathbb{R}^d \to \mathbb{R}^m$, where $f_\theta(x)$ predicts the signal value at location $x \in \mathbb{R}^d$. The parameters $\theta$ are learned from a set of observations $\{(x_i, z_i)\}_{i=1}^N$. This formulation allows querying the signal at arbitrary locations, which is essential for tasks such as interpolation, super-resolution, and inverse problems. Only limited research has explored the use of INRs within generative frameworks, for domain-agnostic architectures (Wang et al., 2025) or surface distribution learning (Ding et al., 2023).

To improve the expressivity of INRs, especially for high-frequency signals, it is common to augment the input coordinates with Random Fourier Features (RFFs) (Rahimi & Recht, 2007). This approach involves projecting the input $x$ into a higher-dimensional space that approximates a Gaussian kernel function with variance $\frac{1}{\sigma_{RFF}^2}$. This choice is motivated by Bochner's theorem, which states that any shift-invariant kernel $K(x, x') = K(x - x')$ on $\mathbb{R}^d$ can be expressed as the Fourier transform of a non-negative measure (Cooper & Bochner, 1957; Puckette & Rudin, 1965). In particular, this implies that a Gaussian kernel $K$ can be approximated via random features sampled from its spectral density[1]:

$$
\gamma(x) = [\cos(Bx), \sin(Bx)]^T
\tag{5}
$$

where $B \in \mathbb{R}^{m \times d} \sim \mathcal{N}(0, \sigma_{RFF}^2 I_m)$. This transformation enables the model to better capture fine-grained variations and non-local dependencies in the signal. Recent works have demonstrated that RFFs significantly enhance the ability of INRs to represent complex spatial structures and generalize from sparse observations (Tancik et al., 2020).

## 3. RP Flow

Let $\mathcal{X} \subset \mathbb{R}^d$ denote a spatial domain (generally $d \in \{1, 2, 3\}$) and let $(\Omega, \mathcal{F}, p)$ be a probability space. Our objective is to construct a function that transports realizations of a source random process defined on $(\mathcal{X}, \Omega)$ to realizations of a target random process, given limited observations of the latter. The general framework of our proposed method, *RP Flow*, is presented in Figure 1. We employ the CFM framework to learn a transport map between the distributions of the source and target processes at each spatial position $x \in \mathcal{X}$. The spatial relationships between positions are represented through an INR. Finally, we use the invertibility of the learned flow to define posterior processes, enabling conditioning on the available observations. Under suitable assumptions on the learned flow, we demonstrate that the transport function preserves both the regularity and statistical properties of the processes.

---

[1]The usual $2\pi$ factor is omitted to maintain consistency with the Gaussian process literature.

## 3.1. Transport of random processes

Let $Z : \mathcal{X} \times \Omega \to \mathbb{R}^m$ be a random process, which we refer to as the *target process*. With a slight abuse of notation, for all $x \in \mathcal{X}$, we denote $Z(x)$ the marginal distribution of $Z$ at position $x$ and for all $\omega \in \Omega$, we denote $Z(\cdot, \omega) = (x \mapsto Z(x, \omega))$ a realization of the random process. In the self-supervised setting that we consider, we have access to a single, possibly noisy, realization $Z(\cdot, \omega_0)$ on a finite set of positions $\bar{X} = \{x_i\}_{i=1}^N \subset \mathcal{X}$.

Let $\xi : \mathcal{X} \times \Omega \to \mathbb{R}^m$ be another random process that we can fully characterize and ideally sample from easily. We refer to it as the *source process*. In this work, we choose to model $\xi$ as a zero-mean Gaussian process with Gaussian covariance $K$, so that the marginal distribution at any given position $x$ follows the standard Gaussian distribution. These choices enable straightforward training procedure using CFM in Section 3.2 and aligns naturally with RFF embeddings discussed in Section 3.3:

$$\xi \sim \mathcal{GP}(0, K), \quad K(x, x') = e^{\frac{-||x - x'||}{2\sigma_\xi^2}}, \; \forall x, x' \in \mathcal{X} \quad (6)$$

We aim to learn a transport map that pushes samples of $\xi$ at any given position $x$ to samples of $Z$ at this position. Formally, we seek a map $T_\theta : \mathbb{R}^m \times \mathcal{X} \mapsto \mathbb{R}^m$, parameterized by a neural network $\theta$, such that for each position $x \in \mathcal{X}$ the prediction of the target distribution at $x$ is obtained by the pushforward of $\xi(x)$: $\hat{Z}(x) \triangleq T_\theta(\cdot, x) \sharp \xi(x)$.

## 3.2. Flow-based parametrization of $T_\theta$

The function $T_\theta$ is learned using the CFM framework, with the conditional probability path defined as a linear interpolation between samples (Albergo & Vanden-Eijnden, 2023; Liu et al., 2023). Specifically, we consider a space-time dependent process $z_t(x)$, defined for $t \in [0, 1]$ and $x \in \mathcal{X}$, such that $z_0(x) \sim \xi(x)$ and $z_1(x) \sim Z(x)$. During training, we randomly sample $x \in \bar{X}, t \in [0, 1]$ and construct the interpolated trajectory as

$$z_t(x) = t z_1(x) + (1 - t) z_0(x) \quad (7)$$

with the corresponding velocity field

$$v_t^{\text{cond}}(z_t(x), x) = z_1(x) - z_0(x) \quad (8)$$

We train the model following stochastic optimization and sample $z_0(x) \sim \xi(x)$ independently for each training sample $z_1(x) \sim Z(x)$. In that specific case, $\mathcal{GP}(0, K)$ reduces to its marginal distribution at position $x$, chosen to be $\mathcal{N}(0, I_m)$, which greatly simplifies the training process. In the case considered in this work, where only a single observation of the target process $Z(\cdot, \omega_0)$ is available, we

---

**Algorithm 1** Posterior sampling from $\hat{Z}^{\text{post}}$

**Inputs:** $\bar{X}, Z(\bar{X}, \omega_0), X^{\text{test}}, v_\theta, \omega$

$\xi(\bar{X}) \leftarrow \Phi_{t_0=1, t_1=0} \left( v_\theta, \bar{X}, Z(\bar{X}, \omega_0) \right)$
  ▷ Solving the ODE in reverse-time to get Gaussian source observations.
$\xi^{\text{post}} \leftarrow \mathcal{GP}(0, \Sigma^{\text{post}}) \mid \left( \bar{X}, \xi(\bar{X}) \right)$
  ▷ Defining the posterior source Gaussian process.
$\xi^{\text{post}}(X^{\text{test}}, \omega) \leftarrow \text{Sample}(\xi^{\text{post}}, X^{\text{test}}, \omega)$
  ▷ Sampling from the posterior at test positions.
$\hat{Z}^{\text{post}}(X^{\text{test}}, \omega) \leftarrow \Phi_{t_0=0, t_1=1} \left( v_\theta, X^{\text{test}}, \xi^{\text{post}}(\mathcal{X}, \omega) \right)$
  ▷ Solving the ODE forward in time from Gaussian posterior samples.

**Return:** $\hat{Z}^{\text{post}}(X^{\text{test}}, \omega)$

---

sample $z_1(x)$ from $Z(x, \omega_0) + \epsilon$, where $\epsilon \sim \mathcal{N}(0, \sigma^2 I_m)$ to model noisy observations. Similarly to GPR, this additive noise term acts as a regularization parameter which leads to a more reliable calibration of the model's uncertainty estimates as shown in an ablation study in Appendix C. This setup enables training with the standard CFM loss described in Equation 3, allowing a single model to learn the marginal distributions of a spatially dependent process across different locations. Although, in principle, $T_\theta$ could be learned using a broad class of Neural ODE or CNF frameworks, the Flow Matching formulation aligns particularly well with our setting and leads to a simple and stable training procedure.

## 3.3. Implicit source representation

The source process $\xi$ is represented implicitly via a RFF embedding of the spatial position $x \in \mathcal{X}$. Given that the source process $\xi$ is a stationary Gaussian Process with kernel $K$, we approximate $K$ using the finite-dimensional RFF embedding $\gamma(x)$ from Equation 5, with frequencies sampled from the spectral density of $K$. The choice of a Gaussian covariance with a fixed lengthscale $\sigma_\xi$ leads to a simple procedure to draw the RFFs from $\mathcal{N}(0, \sigma_{RFF}^2 = \frac{1}{\sigma_\xi^2})$. In Appendix B.1, we illustrate that this implicit representation converges at $t = 0$ to samples from $\xi$.

By decoupling the spatial dependence from the marginal distribution of the processes, RFFs enable to exploit the simple structure of $\xi(x)$ while implicitly embedding spatial covariance. This approach offers flexibility, as it imposes no spatial regularity requirements on the training points and allows sampling from the GP exclusively at generation time, where a realization of $\xi \sim \mathcal{GP}(0, K)$ is drawn and transported from $t = 0$ to $t = 1$ by integration of the Flow Matching ODE governed by the neural vector field $v_\theta$.

## 3.4. Posterior source and target processes

The training procedure described in the previous sections

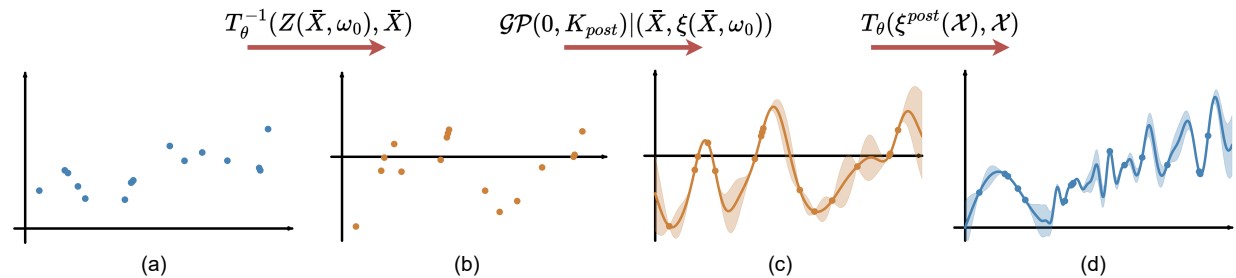

*Figure 2.* Posterior sampling pipeline of RP Flow. The target observations are first projected to the source domain via reverse ODE integration *((a)→(b))*. The created Gaussian source realizations are then interpolated using Gaussian Process regression *((b)→(c))*. The realizations from this posterior GP are finally mapped back to the target space through forward ODE integration *((c)→(d))*.

constitutes a transport $T_\theta$ between prior processes $\xi$ and $\hat{Z}$, containing the assumptions on spatial covariance, regularity, and uncertainty over the measurements. To maximize reconstruction quality, we then want to predict conditionally to the observations $(\bar{X}, Z(\bar{X}))$ to enable access to an unbiased Bayesian posterior. To this purpose, we propose to use the Gaussian properties of the source space as a proxy for the creation of a posterior target. For all $x \in \mathcal{X}, T_\theta(\cdot, x)$ is a diffeomorphism with inverse $T_\theta^{-1}(\cdot, x)$ defined as the integration of the ODE in reverse time. We can then define source observations $\xi(\bar{X}, \omega_0) \triangleq T_\theta^{-1}(Z(\bar{X}, \omega_0), \bar{X})$ as the inverse transport of the target observations $Z(\bar{X}, \omega_0)$. The source space is by construction a Gaussian Process since every finite collection of observation is Gaussian, making it a perfect space for Gaussian Process Regression. We construct the source posterior as a posterior GP $\xi^{post}$, conditioned on the observations $\xi(\bar{X}, \omega_0)$. We discuss the selection of the kernel used for the regression in Appendix C. Finally, a realization of the posterior target process is the transport by $T_\theta$ of a realization of the posterior source process: $\hat{Z}^{post}(\cdot, \omega) \triangleq T_\theta(\xi^{post}(\cdot, \omega), \cdot)$. Sampling from the posterior Gaussian process yields an unbiased target posterior process:

**Proposition 3.1.** *The target posterior process $\hat{Z}^{post}$ is unbiased and verifies $\forall x \in \bar{X}, \omega \in \Omega$,*

$$\hat{Z}^{post}(x, \omega) = Z(x, \omega_0) \qquad (9)$$

In Algorithm 1, we show the detailed procedure for posterior sampling and in Figure 2, we display the main steps in the construction of the posterior processes. The source process has standard Gaussian marginals at each position. In the multivariate case, this property can significantly reduce the computational cost of computing the posterior source process compared to applying GPR directly in the target space. Specifically, due to the independence between source variables, it is possible to compute a posterior GP for each variable individually. This reduces the spatial complexity of the posterior GP from $\mathcal{O}(n^3)$ to $\mathcal{O}(n)$, and the temporal complexity from $\mathcal{O}(n^2)$ to $\mathcal{O}(n)$, for a process with $n$

variables, while preserving the original Gaussian process complexity with respect to the number of evaluated locations. An analysis of the method complexity is provided in Appendix B.4.

### 3.5. Properties

In this section, we analyze key properties of the proposed method to ensure its suitability for generative interpolation in a Neural Implicit framework. Specifically, we aim to embed some desired properties into the source process to be transposed to the target. We suppose that $\forall t \in [0, 1], \forall x \in \mathcal{X}$, both the true conditional velocity $v_t^{cond}(\cdot, x)$ and the learned one $v_\theta(\cdot, x, t)$ are $L_t$-Lipschitz continuous in $t$ for a certain constant $L_t$. Previous work from Benton et al. (2024) shows that under bounded source and target distributions, $v_t^{cond}(\cdot, x)$ is Lipschitz, and in practice $v_\theta(\cdot, x, t)$ is parametrized by a neural network which is a composed of Lipschitz operators and is therefore Lipschitz. The various proofs for this section are developed in Appendix A.

**Lemma 3.2.** *(Lions & Seeger, 2024) Let $x \in \mathcal{X}, z_0(x) \sim \xi(x), z_1(x) \sim Z(x)$. Let $T$ be the function that sends $z_0(x)$ to $z_1(x)$ following the ODE $dz_t(x) = v_t(z_t(x), x)dt$. Suppose that $\forall t \in [0, 1], v_t$ is $L_t$-Lipschitz continuous in $z_t(x)$. Then $T$ is Lipschitz continuous in $z_0(x)$ with constant*

$$L \le \exp\left(\int_0^1 L_t \, dt\right) \qquad (10)$$

This lemma 3.2 guarantees that it suffices for $v_\theta$ to be Lipschitz continuous, rather than requiring the same property for $T_\theta$.

**Theorem 3.3.** *If $\forall x \in \mathcal{X}, \forall t \in [0, 1], v_\theta$ is continuous in $x$ and $t$, then the following regularity properties of the stochastic processes are preserved by the transport map $T_\theta$:*

1. *If $\forall x \in \mathcal{X}, \forall t \in [0, 1], v_\theta$ is continuous in $z_t(x)$, then $\xi$ has almost surely continuous sample paths if and only if $\hat{Z}$ also has almost surely continuous sample paths.*

2. *If $\forall x \in \mathcal{X}, \forall t \in [0, 1]$, $v_\theta$ is Lipschitz continuous in $z_t(x)$, then $\xi$ is mean-square continuous if and only if $\hat{Z}$ is mean-square continuous.*

3. *If $\forall t \in [0, 1], v_\theta(\cdot, \cdot, t) \in C^k(\mathbb{R}^m \times \mathcal{X}, \mathbb{R}^m)$, then $\xi$ has sample paths in $C^k(\mathcal{X}, \mathbb{R}^m)$ if and only if $\hat{Z}$ has sample paths in $C^k(\mathcal{X}, \mathbb{R}^m)$.*

For specific problems, prior knowledge about the regularity of the target field may be available. Theorem 3.3 guarantees that the regularity of the source process is transferred to the target process, which can lead to design choices aimed at improving predictive performance. These equivalences allow RP Flow to explicitly model discontinuities within the source space. This indicates a substantial improvement over classical INRs which are typically bound to model continuous signals. In Appendix B.2, we empirically show this implication of the theorem on a 1D discontinuous signal.

**Theorem 3.4.** *Let $k \geq 1$. If $\forall \omega \in \Omega, \forall x \in \mathcal{X}$, $T_\theta$ is $L$-Lipschitz continuous in $\xi(x, \omega)$, and if $\xi(x)$ has finite centered moments up to order $k$, then $\forall x \in \mathcal{X}$:*

$$\|\mathbb{E}[T_\theta(\xi(x), x] - T_\theta(\mathbb{E}[\xi(x)], x)\| \leq$$
$$L \cdot \left(\mathbb{E}\left[\|\xi(x) - \mathbb{E}[\xi(x)]\|^k\right]\right)^{1/k} \quad (11)$$

*and*

$$\mathbb{E}\left[\|T_\theta(\xi(x), x) - \mathbb{E}[T_\theta(\xi(x), x)]\|^k\right] \leq$$
$$(2L)^k \cdot \mathbb{E}\left[\|\xi(x) - \mathbb{E}[\xi(x)]\|^k\right] \quad (12)$$

RP Flow belongs to the class of ensemble methods, as we only have access to samples from $\hat{Z}$ and $\hat{Z}^{\text{post}}$. A limitation of our approach is the lack of direct access to the statistics of the constructed target process. However, Theorem 3.4 guarantees that the mean and higher-order moments of the target process at each location remain close to those of the source process. This result leads to convergence rates for Monte Carlo approximations of these statistics, as discussed in Appendix B.3. In particular, for the posterior target process, the theorem induces a reduced variance in regions closer to the observations. However, without control on the Lipschitz constant, these bounds may still be large.

# 4. Experiments

We evaluate RP Flow on two challenging spatial interpolation problems: image regression and seismic data interpolation. The first experiment demonstrates the capability of the proposed method to generate high-quality fields as well as predict calibrated uncertainties, and the second reveals the ability to generate high-quality samples in high dimension when data sparsity is greater. We evaluate reconstruction quality using the Peak Signal-to-Noise Ratio (PSNR) and

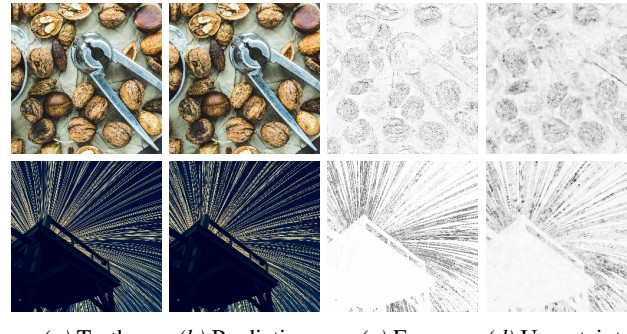

| *(a)* Truth | *(b)* Prediction | *(c)* Error | *(d)* Uncertainty |

*Figure 3.* Qualitative results of RP Flow for the image regression ($4\times$ upsampling) task. The posterior process $\hat{Z}^{post}$ enables high-quality predictions *(b)*. The per-pixel predictive distribution $\hat{Z}(x)$ allows for calibrated uncertainty quantification *(d)*.

the Structural Similarity Index Metric (SSIM), sample quality using the Wasserstein distance ($\mathcal{W}_1$) and calibration with the Probabilistic Calibration Error ($PCE_1$):

$$PCE_p(\hat{Z}) = \int_0^1 \left|\hat{c}_t(\hat{Z}) - c_t\right|^p dt \quad (13)$$

where $c_t$ is the confidence interval of order $t$, and $\hat{c}_t(\hat{Z})$ is the empirical coverage of this interval, based on the ensemble prediction $\hat{Z}$. As we can only sample from $\hat{Z}$, we need to rely on a Monte Carlo approximation of $PCE_p$.

This metric gives a quantitative value linking confidence sets with uncertainty: A 90% prediction interval contains the true realization 90% of the time (Dheur & Taieb, 2023; Zhou et al., 2021).

For each task, we compare RP Flow with INR and Bayesian baselines. Specifically, we consider a Random Fourier Feature Network (Puckette & Rudin, 1965), a SIREN Network (Sitzmann et al., 2020), two Gaussian Process Regressors: noiseless to maximize PSNR and calibrated to minimize PCE (via additive noise $\sigma$ in Equation 4) and a Deep Gaussian Process (Damianou & Lawrence, 2013).

## 4.1. Image regression

To demonstrate the properties of our method, we evaluate RP Flow on two image regression tasks using the Div2K dataset (Timofte et al., 2017). Specifically, we consider the $4\times$ upsampling problem and the reconstruction from 25% randomly sampled pixels from the image, on a subset of 32 images from the dataset. For each image, the models are trained on the 25% training pixels and are evaluated on all the remaining ones. All models use a Gaussian kernel for spatial covariance (explicitly or via RFFs). For RP Flow, we report the reconstruction results (PSNR and SSIM) on samples from the posterior process, as well as the calibration results (PCE) of the prior process. All models are tuned

| METHOD | PSNR↑ | SSIM↑ | PCE$_1$↓ |
|---|---|---|---|
| **UPSAMPLING** | | | |
| RFF NETWORK | 25.20±4.93 | **0.82±0.08** | |
| SIREN | 24.64±4.87 | 0.81±0.09 | |
| GPR$_{\text{NOISELESS}}$ | 23.57±4.17 | 0.79±0.09 | 0.39±0.05 |
| GPR$_{\text{CALIBRATED}}$ | 22.42±3.09 | 0.58±0.07 | 0.12±0.05 |
| DEEP GP | 19.09±2.93 | 0.36±0.11 | 0.35±0.04 |
| RP FLOW | **25.30±5.05** | **0.82±0.08** | **0.09±0.05** |
| **RANDOM** | | | |
| RFF NETWORK | 23.67±4.66 | **0.78±0.09** | |
| SIREN | 22.19±4.29 | 0.72±0.11 | |
| GPR$_{\text{NOISELESS}}$ | 17.94±3.69 | 0.58±0.11 | 0.41±0.04 |
| GPR$_{\text{CALIBRATED}}$ | 20.36±2.70 | 0.49±0.06 | 0.13±0.05 |
| DEEP GP | 19.08±2.94 | 0.37±0.11 | 0.36±0.04 |
| RP FLOW | **23.70±4.64** | 0.76±0.09 | **0.09±0.04** |

*Table 1.* Image Regression results on the Div2K dataset. All results are given in PSNR, SSIM and PCE. Results are shown in mean ± standard deviation. "Upsampling" represents the 4× upsampling task and "Random" represents the scenario where 25% of the pixels are randomly sampled for training.

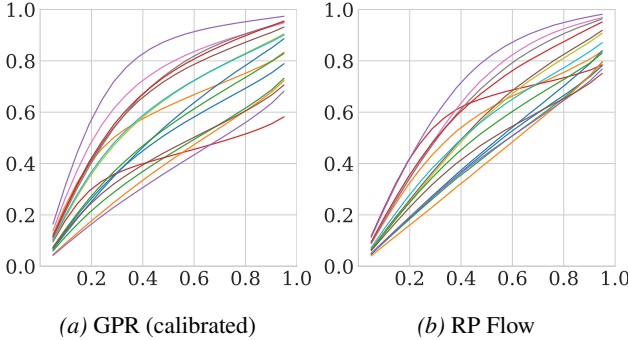

*(a)* GPR (calibrated)        *(b)* RP Flow

*Figure 4.* Reliability diagrams for RP Flow and calibrated GPR on the 16 test images in the 4× upsampling setting. For each confidence level (shown on the $x$-axis), the corresponding empirical coverage achieved by the model is reported on the $y$-axis. Although the PCE results of both methods are close in expectation (Table 1), RP Flow demonstrates greater consistency in per-image calibration.

to achieve the best performance on 16 images and results are reported on the remaining test images. Additional results, including an ablation study of various parameters, are provided in Appendix C.

Figure 3 illustrates the performance of RP Flow on two test images from the dataset. The posterior process produces high-quality reconstructed samples, while the ensemble predictions provide calibrated uncertainty estimates. In particular, RP Flow induces higher uncertainty in high-frequency areas. Table 1 shows that the proposed method achieves comparable results to the discriminative INRs in both upsampling and random scenarios while the access to a predictive distribution enables our model to capture uncertainty similarly to Gaussian Processes. Notably, at optimal calibration, the predictive variance of our method ($0.08 \pm 0.01$) is smaller than that of the GPR ($0.16 \pm 0.00$), illustrating reduced epistemic uncertainty. In this work, we focus on evaluating the model's calibration without applying any post-hoc recalibration, a setting that makes it difficult to obtain consistent reliability diagrams across a diverse high-frequency dataset (Figure 4).

### 4.2. Seismic interpolation

Seismic imaging is a key technique in geophysics used to infer the structure of the Earth's subsurface. It involves sending acoustic waves into the ground and recording the reflected signals (called seismic traces) at various spatial locations resulting, after processing, in a 3D volume of data. Each vertical trace captures a complex, location-dependent signal that constitutes a high-dimensional sample. In practice, seismic data is often sparsely sampled on a regular 2D

grid, while the goal is to reconstruct the dense 3D volume of traces. This task, known as seismic interpolation, is challenging because the distribution of traces varies significantly across space, and the underlying process exhibits strong spatial correlations with often the impossibility to access 3D data for supervision. To address this task, Gaussian Process regression, referred to as Kriging in the geostatistics literature (Christianson et al., 2023), is usually employed to provide both an estimation of the seismic data and an associated uncertainty at each location. The recent application of seismic data in emerging contexts, such as offshore wind farm development, has highlighted limitations in the use of Kriging. These limitations become particularly evident as the sparsity of 2D measurement grid increases. This setting is however interesting to demonstrate the capabilities of RP Flow in a high-dimensional, structured setting with sparsely sampled training points.

In this section, we parametrize RP Flow by position-conditioned U-Nets (Ronneberger et al., 2015; Dhariwal & Nichol, 2021) with different sizes and compare them against the different baselines. Although 3D generative models could serve as baselines, they require numerous seismic volumes for training, which is impractical given the uniqueness of each volume. Therefore, we excluded them from our experiments and instead focus on comparing RP Flow exclusively against transductive methods. We empirically demonstrate that our model outperforms GPs and INRs in reconstruction quality (PSNR, SSIM), trace sample quality ($\mathcal{W}_1$) and uncertainty calibration (PCE$_1$). In this experiment, we utilize two datasets. First, we investigate the properties of RP Flow in a controlled setup and create a synthetic dataset of 32 seismic cubes using similar simulation properties (Merrifield et al., 2022). Second, to demonstrate the capabilities of the model on real-world noisy data, we

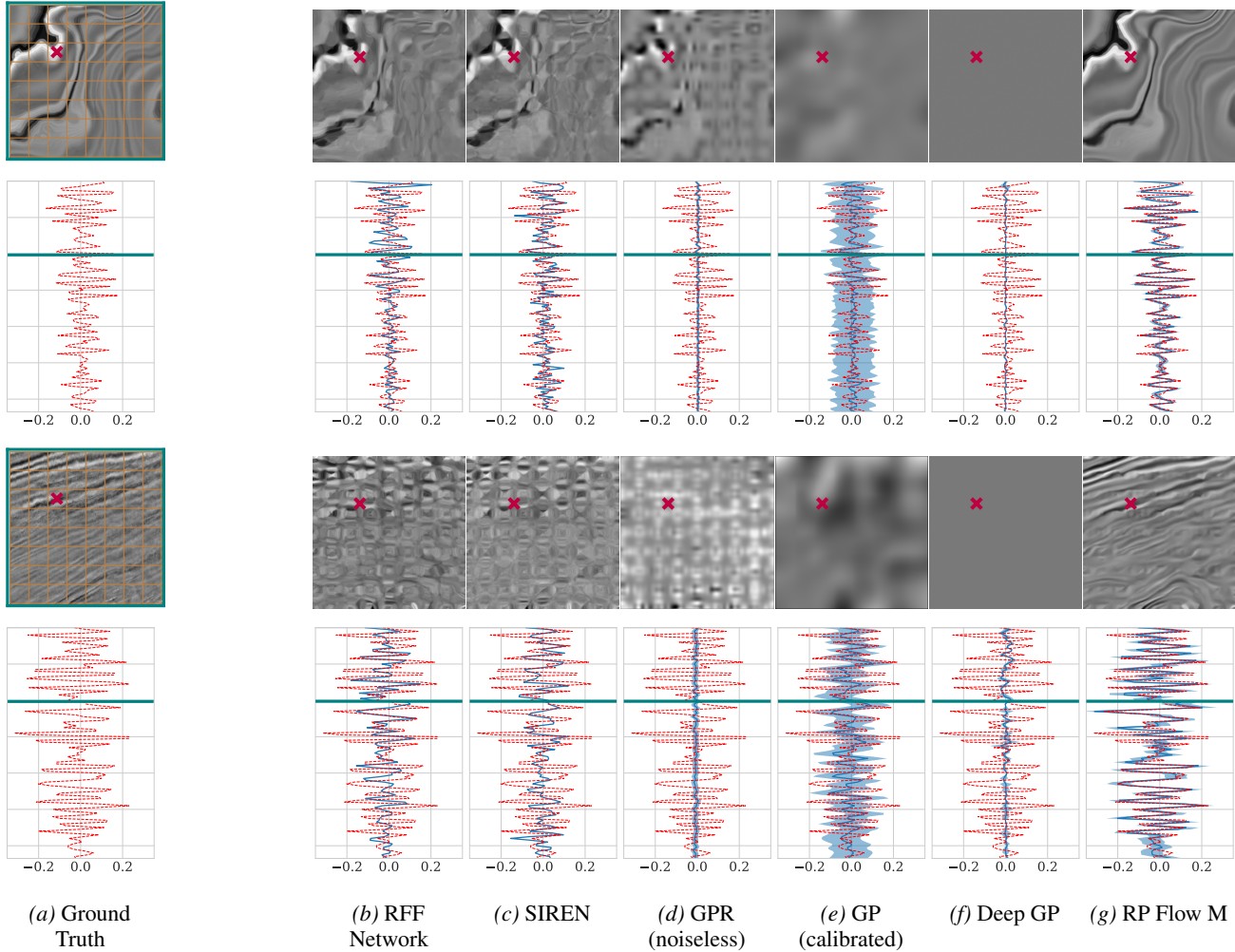

*(a) Ground Truth*  *(b) RFF Network*  *(c) SIREN*  *(d) GPR (noiseless)*  *(e) GP (calibrated)*  *(f) Deep GP*  *(g) RP Flow M*

*Figure 5.* Qualitative results on a synthetic and real seismic volume. **Top:** Horizontal slice of the 3D volume predicted by the different models. **Bottom:** Predictive distribution of a vertical trace by the different models at a single test position. Ground truth trace is represented in dashed red, mean prediction in solid blue and standard deviation in light blue. On the Ground Truth *(a)*, the **orange** grid denotes the training positions. The **red** cross on the slices indicate the location of the traces shown below. The **green** line on the traces correspond to the depth of the slices displayed above.

use the F3 open source dataset (Silva et al., 2019). Each volume is cropped to $513 \times 513 \times 128$. For training, we consider a sparse grid of traces where the resulting 2D seismic sections are spaced 64 units apart in both directions. Additional settings such as irregular grid and randomly sampled training traces are discussed in Appendix D along with the implementation details for these experiments.

We report all quantitative results in Table 2. In both the synthetic and real cases, every instance of RP Flow outperforms the baselines across all metrics, demonstrating superior reconstruction quality, sample quality and calibration. Notably, increasing the model size improves performance up to a certain point, beyond which the limited amount of available data constrains further learning. However, the continuity of the positional embedding serves as a regularization

mechanism, mitigating overfitting in larger models.

Figure 5 displays visual results of the models on both the synthetic and real datasets. We present a two-dimensional horizontal slice of the 3D predicted volumes reconstructed by the different models as well as the predictive distributions of the different models against a single trace extracted from the seismic volumes. Visually, RP Flow is the only model capable to interpolate accurately between the training traces, while providing calibrated and reduced uncertainty without losing the high-frequency complex patterns in the signal. More generally, this experiment demonstrates that the RP Flow framework performs effectively in high-dimensional settings where the sample structure is complex, yet the random process exhibits strong spatial correlation.

| | SYNTHETIC | | | | REAL | | | |
|---|---|---|---|---|---|---|---|---|
| METHOD | PSNR↑ | SSIM↑ | $\mathcal{W}_1 \downarrow$ | PCE↓ | PSNR↑ | SSIM↑ | $\mathcal{W}_1 \downarrow$ | PCE↓ |
| RFF NETWORK | 26.78±1.78 | 0.75±0.09 | 0.35±0.07 | | 26.11 | 0.65 | 0.47 | |
| SIREN | 27.05±1.54 | 0.73±0.07 | 0.36±0.07 | | 26.29 | 0.65 | 0.47 | |
| GPR (NOISELESS) | 26.56±1.64 | 0.73±0.09 | 0.42±0.10 | 0.38±0.08 | 25.78 | 0.59 | 0.51 | 0.47 |
| GPR (CALIBRATED) | 26.22±1.74 | 0.62±0.11 | 0.50±0.10 | 0.13±0.07 | 25.13 | 0.48 | 0.56 | 0.06 |
| DEEP GP | 26.10±1.63 | 0.60±0.11 | 0.54±0.10 | 0.40±0.01 | 24.60 | 0.38 | 0.66 | 0.48 |
| RP FLOW S ($\#params = 1.7M$) | 31.24±1.70 | 0.91±0.04 | 0.39±0.08 | **0.06±0.03** | 28.47 | 0.78 | 0.40 | **0.01** |
| RP FLOW M ($\#params = 10M$) | **32.08±1.73** | **0.93±0.04** | **0.33±0.05** | **0.06±0.04** | **28.77** | **0.80** | **0.39** | 0.02 |
| RP FLOW L ($\#params = 50M$) | 31.84±1.76 | 0.92±0.04 | 0.34±0.06 | 0.07±0.03 | 28.63 | 0.79 | **0.39** | 0.04 |

*Table 2.* Quantitative results of the seismic interpolation task on the two datasets. For the synthetic dataset, results are reported as mean ± standard deviation over 16 evaluation volumes, whereas for the real seismic dataset, the single scalar metric is provided. In both cases, each instance of RP Flow consistently outperforms the INR and Gaussian Processes baselines in terms of reconstruction quality, sample quality, and uncertainty calibration. Increasing the model size improves performance up to a point, beyond which the limited data quantity becomes a bottleneck for larger models.

## 5. Conclusions

In this work we introduced RP Flow, a neural implicit generative model that uses Flow Matching to model the marginal distributions of a random process at each spatial position. Our approach employs an Implicit Neural Representation to model the transport between the random processes, enabling learning from a single realization of the process at observed locations. We exploited the Gaussian properties of the source space to define posterior processes via Gaussian Process regression between source observations. The theoretical foundations of our method allow for modeling of various regularity assumptions on the process through the source process definition and ensure robust convergence of target statistics via ensemble sampling.

Our experimental results on both image regression and seismic interpolation demonstrates strong performances of RP Flow in terms of data reconstruction, sample quality and uncertainty calibration. In high-frequencies settings, we achieved comparable reconstruction results as the INR baselines while providing superior calibration compared to GPs. In high-dimensional settings, RP Flow outperforms the GP and INR baselines, achieving state-of-the-art results in sparse seismic interpolation without relying on external supervision. In conclusions, RP Flow represents a promising step towards generative models for reconstruction problems in which calibrated uncertainty must be quantified and additional data cannot be acquired.

**Limitations and Future work.** While RP Flow demonstrates strong performance in terms of reconstruction quality and uncertainty calibration, the requirement to compute a posterior Gaussian process constitutes a significant computational bottleneck. Future work could investigate sparse GP approximations to mitigate this cost. Additionally, extending RP Flow to indirect supervised tasks represents a promising direction, as does the application of meta-learning approaches to learn a prior across multiple RP Flows.

## Acknowledgments

We thank the anonymous reviewers for their valuable feedback. We acknowledge TotalEnergies for supporting this work and for permitting the publication of these results. Any opinions, findings, and conclusions expressed in this paper are those of the authors and do not necessarily reflect the views of TotalEnergies. This work was supported by the French National Association for Research and Technology (ANRT) through a CIFRE fellowship.

## Impact Statement

This paper addresses generative models for interpolation problems. While we expect limited societal impact from our contribution, risks can arise from potential over-reliance on interpolated outputs. Users should treat results as probabilistic estimates rather than ground truth. Additionally, users should weigh the environmental and computational cost of GPU-intensive inference against the practical benefits gained from applying the proposed method.

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

# A. Proofs

*Proof.* Proposition 3.1.

Let $x \in \bar{X}$ and $\omega \in \Omega$. We have:

$$\hat{Z}^{\text{post}}(x, \omega) \triangleq T_\theta \left( x, \xi^{\text{post}}(x, \omega) \right) \tag{14}$$

We define $\xi^{\text{post}}$ as a posterior Gaussian Process, conditioned on $\left( \bar{X}, T_\theta^{-1}(\bar{X}, Z(\bar{X}, \omega)) \right)$, defined in Equation 4 in the noiseless case. Therefore, since the variance at conditioning positions is zero we have:

$$\xi^{\text{post}}(x, \omega) = T_\theta^{-1}(x, Z(x, \omega_0)) \tag{15}$$

Then,

$$\begin{aligned} \hat{Z}^{\text{post}}(x, \omega) &= T_\theta \left( x, T_\theta^{-1}(x, Z(x, \omega_0)) \right) \\ &= Z(x, \omega_0) \end{aligned} \tag{16}$$

Finally, $\hat{Z}^{\text{post}}$ is unbiased. $\qquad \square$

*Proof.* Lemma 3.2 (Lions & Seeger, 2024).

Let $z_t^i$ and $z_t^j$ be two solutions at time $t$ of the ODE with initial conditions $z_0^i$ and $z_0^j$. Let

$$\delta_t \triangleq \| z_t^i - z_t^j \| \tag{17}$$

Using the triangle inequality for the differentiation of the norm, we have

$$\begin{aligned} \frac{d}{dt} \delta_t &= \frac{d}{dt} \| z_t^i - z_t^j \| \\ &\leq \| v_t(z_t^i) - v_t(z_t^j) \| \end{aligned} \tag{18}$$

Then, by applying the Lipschitz constraint on $v_t$,

$$\begin{aligned} \frac{d}{dt} \delta_t &\leq L_t \| z_t^i - z_t^j \| \\ &\leq L_t \delta_t \end{aligned} \tag{19}$$

Finally, by Grönwall's inequality,

$$\delta_t \leq \delta_0 \exp \left( \int_0^t L_s \, ds \right) \tag{20}$$

Evaluating at $t = 1$ gives

$$\| T(z_0^i) - T(z_0^j) \| = \delta_1 \leq \delta_0 \exp \left( \int_0^1 L_s \, ds \right) \tag{21}$$

Therefore, $T$ is Lipschitz with constant $L \leq \exp \left( \int_0^1 L_t \, dt \right)$. $\qquad \square$

*Proof.* Theorem 3.3.

Let $T_\theta$ be the transport map defined as in Lemma 3.2. By assumption, for all $t \in [0, 1]$ and $x \in \mathcal{X}$, the vector field $v_\theta$ is continuous in $(x, t)$. Then, $T_\theta$ is continuous in $x$.

Consider the stochastic process $\xi : \mathcal{X} \times \Omega \to \mathbb{R}^m$ and define $\hat{Z}(x, \omega) \triangleq T_\theta(\xi(x, \omega), x)$. We check the forward implication of each property:

**Almost sure continuity of sample paths.** Suppose that $\forall x \in \mathcal{X}, \forall t \in [0,1]$, $v_\theta$ is continuous in $z_t(x)$. Then $\forall x \in \mathcal{X}, T_\theta$ is continuous in $\xi(x)$. If $\xi$ has almost surely continuous sample paths, then for almost every $\omega$, the map $x \mapsto \xi(x, \omega)$ is continuous. Since $T_\theta$ is continuous in $(z, x)$, the composition $x \mapsto T_\theta(\xi(x, \omega), x)$ is also continuous. Hence, $\hat{Z}$ has almost surely continuous sample paths. The equivalence can be established in the same manner using the inverse transport $T_\theta^{-1}$.

**Mean-square continuity.** Suppose that $\forall x \in \mathcal{X}, \forall t \in [0,1]$, $v_\theta$ is Lipschitz continuous in $z_t(x)$. Then from Lemma 3.2, $T_\theta$ is Lipschitz continuous in $z$.

Assume $\xi$ is mean-square continuous, i.e.,

$$\lim_{x' \to x} \mathbb{E}\big[\|\xi(x) - \xi(x')\|^2\big] = 0 \tag{22}$$

Since $T_\theta$ is Lipschitz in $z$ with constant $L$ (Lemma 3.2), we have

$$\begin{aligned}
\|\hat{Z}(x) - \hat{Z}(x')\| &= \|T_\theta(\xi(x)) - T_\theta(\xi(x'))\| \\
&\leq L\|\xi(x) - \xi(x')\|
\end{aligned} \tag{23}$$

Squaring and taking expectation gives

$$\begin{aligned}
\mathbb{E}\big[\|\hat{Z}(x) - \hat{Z}(x')\|^2\big] &\leq L^2 \mathbb{E}\big[\|\xi(x) - \xi(x')\|^2\big] \\
&\xrightarrow[x' \to x]{} 0
\end{aligned} \tag{24}$$

so $\hat{Z}$ is mean-square continuous.

The equivalence can be established in the same manner using the inverse transport $T_\theta^{-1}$.

$C^k$ **regularity of sample paths.** Suppose that $\forall t \in [0,1]$, $v_\theta(\cdot, \cdot, t) \in C^k(\mathbb{R}^m \times \mathcal{X}, \mathbb{R}^m)$. Then $T_\theta \in C^k(\mathbb{R}^m \times \mathcal{X}, \mathbb{R}^m)$. If $\xi$ has sample paths in $C^k(\mathcal{X}, \mathbb{R}^m)$, then for almost every $\omega$, the composition

$$x \mapsto T_\theta(\xi(x, \omega), x) \tag{25}$$

is $C^k$ by the chain rule for multivariate functions. Hence, $\hat{Z}$ has sample paths in $C^k(\mathcal{X}, \mathbb{R}^m)$.

The equivalence can be established in the same manner using the inverse transport $T_\theta^{-1}$. $\qquad\square$

*Proof.* Theorem 3.4.
Assume $T$ is $L$-Lipschitz. Let $x \in \mathcal{X}$. $\xi(x)$ is a random variable with finite moments up to order $k \geq 1$. Let $\mu(x) = \mathbb{E}[\xi(x)]$.

**First inequality.** We have:

$$\begin{aligned}
\|\mathbb{E}[T(\xi(x), x)] - T(\mu(x))\| &= \|\mathbb{E}[T(\xi(x), x) - T(\mu(x))]\| \\
&\leq \mathbb{E}\big[\|T(\xi(x), x) - T(\mu(x))\|\big]
\end{aligned} \tag{26}$$

Since $T$ is $L$-Lipschitz in $\xi(x, \omega)$, $\forall \omega \in \Omega$,

$$\|T(\xi(x, \omega), x) - T(\mu(x))\| \leq L\|\xi(x, \omega) - \mu(x)\| \tag{27}$$

Thus,

$$\|\mathbb{E}[T(\xi(x), x)] - T(\mu(x))\| \leq L\, \mathbb{E}\big[\|\xi(x) - \mu(x)\|\big] \tag{28}$$

Applying Hölder's inequality with exponent $k$ gives

$$\mathbb{E}\big[\|\xi(x) - \mu(x)\|\big] \leq \big(\mathbb{E}\big[\|\xi(x) - \mu(x)\|^k\big]\big)^{1/k} \tag{29}$$

Therefore,

$$\|\mathbb{E}[T(\xi(x), x)] - T(\mu(x))\| \le L \cdot \left(\mathbb{E}\left[\|\xi(x) - \mu(x)\|^k\right]\right)^{1/k} \tag{30}$$

**Second inequality.** Let $\omega \in \Omega$. We have:

$$\|T(\xi(x,\omega), x) - \mathbb{E}[T(\xi(x), x)]\| = \|T(\xi(x,\omega), x) - T(\mu(x)) + T(\mu(x)) - \mathbb{E}[T(\xi(x), x)]\|$$
$$\le \|T(\xi(x,\omega), x) - T(\mu(x))\| + \|T(\mu(x)) - \mathbb{E}[T(\xi(x), x)]\| \tag{31}$$

Raising to the power $k$ and using that $|a + b|^k \le 2^{k-1}(|a|^k + |b|^k)$, we get

$$\|T(\xi(x,\omega), x) - \mathbb{E}[T(\xi(x), x)]\|^k \le 2^{k-1}\left(\|T(\xi(x,\omega), x) - T(\mu(x))\|^k + \|T(\mu(x)) - \mathbb{E}[T(\xi(x), x)]\|^k\right) \tag{32}$$

Using Jensen's inequality, we have

$$\|T(\mu(x)) - \mathbb{E}[T(\xi(x), x)]\|^k = \|\mathbb{E}[T(\mu(x)) - T(\xi(x), x)]\|^k \tag{33}$$
$$\le \left(\mathbb{E}[\|T(\mu(x)) - T(\xi(x), x)\|]\right)^k \tag{34}$$
$$\le \mathbb{E}[\|T(\mu(x)) - T(\xi(x), x)\|^k] \tag{35}$$

Plugging this into Equation 32 and taking expectation yields

$$\mathbb{E}\left[\|T(\xi(x,\omega), x) - \mathbb{E}[T(\xi(x), x)]\|^k\right] \le 2^k \mathbb{E}\left[\|T(\xi(x,\omega), x) - T(\mu(x))\|^k\right] \tag{36}$$

Finally, using the Lipschitz property,

$$\mathbb{E}\left[\|T(\xi(x,\omega), x) - T(\mu(x))\|^k\right] \le (2L)^k \mathbb{E}\left[\|\xi(x,\omega) - \mu(x)\|^k\right] \tag{37}$$

$\square$

# B. Properties of RP Flow

In this section, we illustrate the properties of the presented method, described in Section 3.5. For this purpose, we consider a 1D stochastic process and parametrize RP Flow with a ReLU MLP. More specifically, we illustrate that in this case the learned function converges implicitly to the right Gaussian Process in the source space, that Theorem 3.3 allows us to model known discontinuities in the target process and that Theorem 3.4 leads to convergence rates for the moments of $\hat{Z}$.

**B.1. Implicit convergence at $t = 0$**

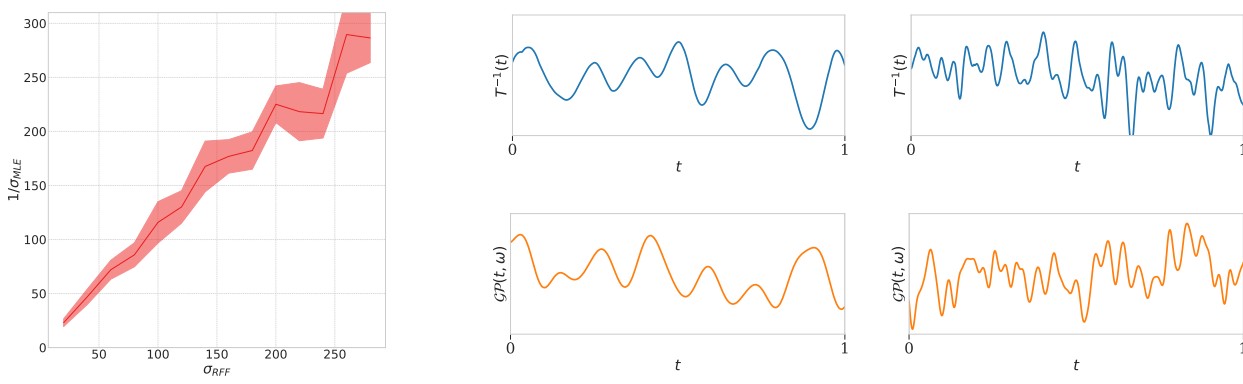

*Figure 6.* RP Flow converges to a sample from a Gaussian Process with Gaussian covariance and lengthscale $\sigma_{GP} = 1/\sigma_{RFF}$ in the case of a constant target process $Z$. Left figure shows the lengthscale $\sigma$ that maximize the likelihood of $T^{-1}(Z)$ to be a sample from $\mathcal{N}(0, \sigma)$, for different values of $\sigma_{RFF}$. Right figure displays samples from $T^{-1}(Z)$ and $\mathcal{N}(0, 1/\sigma_{RFF})$ for $\sigma_{RFF} = 20$ and $\sigma_{RFF} = 80$.

In Section 3.3, the source process is not explicitly provided to the model during training. Instead, the source samples $\xi(x)$ are drawn from a standard Gaussian distribution. However, by definition, the source space corresponds to a Gaussian process, and Bochner's theorem provides intuition regarding its covariance function. Since the position $x$ is introduced to the model through Random Fourier Features, we hypothesize that the source process should exhibit a covariance function associated with the position embedding $B$, as defined in Section 2.

We validate that in a simple 1D case. For this purpose, we consider the case where the target process $Z$ is constant equal to zero: $\forall x \in \mathcal{X}, \forall \omega \in \Omega, Z(x, \omega) = 0$. For different values of $\sigma_{RFF}$, we analyze $T_\theta^{-1}(Z)$ after training. Figure 6 displays the lengthscale of a Gaussian covariance $K_{\sigma_{GP}}$ that maximize the likelihood of $T_\theta^{-1}(Z)$ to be drawn from $\mathcal{N}(0, K_{\sigma_{GP}})$ on the left as well as samples from $T^{-1}(Z)$ and $\mathcal{N}(0, 1/\sigma_{RFF})$ for different values of $\sigma_{RFF}$ on the right. The experiment is ran 64 times per lengthscale value across a wide range of $\sigma_{RFF}$ and systematically, the models converge to samples from a Gaussian process with the correct covariance.

### B.2. Modeling discontinuities

Generally, the parametrization of INRs by continuous neural networks bounds them to represent continuous signals. However, Theorem 3.3 demonstrates that RP Flow transports the regularity of the source process in the target space. In particular, by designing a source process that exhibits a clear discontinuity (either with the almost-surely or the mean-square definition), we can model prior assumptions on the continuity of the target process to learn.

In contrast to other experiments, where we modeled $\xi$ as a Gaussian process with almost-surely continuous sample paths, we now compare the performance of two design choices for $\xi$. The first model is trained and evaluated using a source process that is almost-surely continuous, while the second is trained using a source process that is almost-surely discontinuous at $t = 0.5$. Both models aim to approximate a target process that is itself almost-surely discontinuous at $t = 0.5$. The results are plotted in Figure 7. At $t = 0.5$, the first model produces a continuous representation of $Z$, which does not reflect the true behavior of the process being modeled. In contrast, the second model correctly exhibits a discontinuity at this point.

This property can be crucial in the modeling of inverse problems where regularity assumptions can be made on the target process such as Tomographic reconstruction with known object boundaries (Schweiger & Arridge, 1999) or Seismic inversion with known body location (Jones & Davison, 2014; Ahmed & Güneyli, 2025).

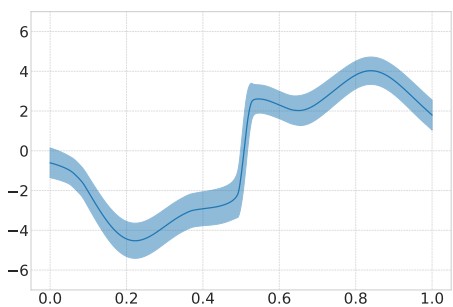
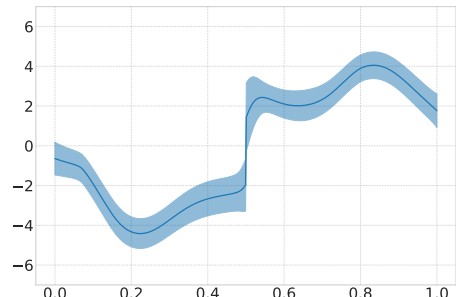

*(a)* Almost-surely continuous source process.

*(b)* Almost-surely discontinuity of the source process at $t = 0.5$.

*Figure 7.* RP Flow transports the regularity of the source process. In the case where the target process $Z$ exhibits a clear discontinuity at $x = 0.5$, we parametrize RP Flow with an almost-surely continuous source process *(a)*, and a source process that is almost-surely discontinuous at $t = 0.5$ *(b)*. The resulting predicted fields have the same regularity as the source parametrization, as ensured by Theorem 3.3.

### B.3. Estimation of mean and higher-order moments

In this section, we examine the implications of Theorem 3.4 for deriving a tail bound on the sample mean estimator, as stated in Proposition B.1.

**Proposition B.1** (Sample mean Tail Bound)**.** *Let $x \in \mathcal{X}$. Let $\xi(x)$ be a random variable in $\mathbb{R}^m$ with finite non-zero variance,*

*and let $T$ be $L$-Lipschitz in $\xi(x)$, in the sense of Theorem 3.4. Define*

$$\mu(x) \triangleq \mathbb{E}[T(\xi(x), x)] \qquad \hat{\mu}_N(x) \triangleq \frac{1}{N} \sum_{i=1}^{N} T(\xi_i(x), x) \tag{38}$$

*where $\xi_1(x), \ldots, \xi_N(x)$ are i.i.d. samples from $\xi(x)$. Then for all $t > 0$,*

$$\mathbb{P}(\|\hat{\mu}_N(x) - \mu(x)\| > t) \leq \frac{\mathrm{Var}(T(\xi(x), x))}{N t^2} \leq \frac{4L^2 \mathrm{Var}(\xi(x))}{N t^2} \tag{39}$$

*Proof.* This proposition is a direct application of Chebyshev's inequality extended from $T(\xi)$ to $\xi$ by Theorem 3.4. □

This result illustrates the type of bounds on Monte Carlo estimators that can be extended from $\xi$ to $T(\xi)$ using Theorem 3.4. It shows that the process generated by RP Flow admits a characterization in terms of ensemble sampling. Similarly, the same type of bound can be derived for the sample moment estimators $\tilde{m}_{k,N}(x) \triangleq \frac{1}{N} \sum_{i=1}^{N} \|T(\xi_i(x), x) - \mu(x)\|^k$ but is harder to achieve in the case of the plug-in moment estimators $\hat{m}_{k,N}(x) \triangleq \frac{1}{N} \sum_{i=1}^{N} \|T(\xi_i(x), x) - \hat{\mu}_N(x)\|^k$. While additional assumptions on $T(\xi)$, such as sub-Gaussian tail behavior, could yield sharper bounds and similar bounds for plug-in moment estimators, a full development of these refinements lies outside the scope of this work. Our aim here is simply to demonstrate that the method already enjoys robust properties in the general setting.

### B.4. Computational complexity

RP Flow requires solving an ODE to transport samples and constructing a posterior Gaussian Process in the source space. Both components are computationally expensive. In this section, we analyze the method's complexity in terms of both temporal and spatial complexity in Table 3, and we report the computation time required for each experiment in Table 4.

We consider a multivariate process with $n$ variables, evaluated in $N = N_{train} + N_{test}$ different locations. We also consider $k$ Euler steps to integrate the Flow Matching ODE. The temporal and spatial complexities of the posterior sampling method are reported in Table 3. In this setting, the $n$ variables of the process are not assumed to be independent. The general computational cost of evaluating a posterior GP then scales as $\mathcal{O}((nN)^3)$ in space and $\mathcal{O}((nN)^2)$ in time. However, when sampling from the RP Flow posterior, the posterior GP is computed in the source space which is by construction independent across variables (e.g. channels of an image, depth slices of a seismic volume). This allows one to consider a separate posterior GP for each variable, thereby reducing the computational cost of posterior GP evaluation in the source space to $\mathcal{O}(nN^3)$ in space and $\mathcal{O}(nN^2)$ in time. Although the method still exhibits cubic memory scaling in space, the per-variable independence in the source space already represents a substantial computational improvement. For instance, for a spatial process with $n = 10^3$ variables (which can occur in multi-spectral imaging or seismic imaging), the computational cost of RP Flow is reduced by a factor of $10^6$.

| | TEMPORAL COMPLEXITY | SPATIAL COMPLEXITY |
|---|---|---|
| GP REGRESSION | $\mathcal{O}((nN)^3)$ | $\mathcal{O}((nN)^2)$ |
| REVERSE ODE INTEGRATION | $\mathcal{O}(kN)$ | $\mathcal{O}(N)$ |
| POSTERIOR SOURCE GP | $\mathcal{O}(nN^3)$ | $\mathcal{O}(nN^2)$ |
| FORWARD ODE INTEGRATION | $\mathcal{O}(kN)$ | $\mathcal{O}(N)$ |
| TOTAL (POSTERIOR SAMPLING) | $\mathcal{O}((k+nN)N^2)$ | $\mathcal{O}(nN^2)$ |

*Table 3.* RP Flow temporal and spatial complexity. The results are reported for a multivariate process with $n$ variables, evaluated in $N = N_{train} + N_{test}$ different locations using $k$ Euler steps to integrate the Flow Matching ODE. The general complexity of Gaussian Process regression as well as the three main steps used to sample from the posterior are displayed.

We also report the empirical time required to compute the experiments presented in the main paper in Table 4. In this setting, we fix $n = 3$ for the image regression task and $n = 128$ for the seismic interpolation task. We consider $N = 512^2$ for images, $N = 513^2$ for seismic volumes and use $k = 100$ Euler integration steps to integrate the ODE (both forward and backward).

Although training is fast in both settings, sampling becomes computationally expensive in the seismic interpolation case. The choice of $k = 100$ is made to reduce approximation errors arising from backward and forward ODE integration. However, $k$ can be reduced in practice, and we observe that $k = 10$ is sufficient to achieve good reconstruction quality, while reducing the overall computation time by a factor 10. One could also consider different $k_{backward}$ and $k_{forward}$, as the backward integration of the ODE is harder but requires only to be done once. More generally, we observe that RP Flow has similar training times than the INRs while retaining the Gaussian Process computation times when sampling. This constitutes a computational bottleneck of our method, which could potentially be solved by considering sparse GP approximations or inducing points. However, such approximations are beyond the scope of this work. Consistent to our experiments, both the GP baselines and the GP component of RP FLOW are assumed to be variable-wise independent in this comparison.

| | IMAGE REGRESSION | SEISMIC INTERPOLATION |
|---|---|---|
| **RFF NETWORK** | | |
| TRAINING | 1:45±0:02 | 4:48±0:02 |
| SAMPLING | 0:03±0:00 | 0:03±0:00 |
| **SIREN** | | |
| TRAINING | 1:48±0:02 | 4:57±0:02 |
| SAMPLING | 0:03±0:00 | 0:04±0:00 |
| **GP REGRESSORS** | | |
| TRAINING | 0:51±0:01 | 1:58±0:02 |
| SAMPLING | 0:10±0:01 | 0:47±0:01 |
| **DEEP GP** | | |
| TRAINING | 14:22±0:01 | 2:38:45±0:12 |
| SAMPLING | 0:06±0:00 | 0:54±0:00 |
| **RP FLOW** | | |
| TRAINING | 1:46±0:02 | 5:53±0:02 |
| PRIOR SAMPLING | 2:11±0:01 | 3:46±0:01 |
| REVERSE ODE INTEGRATION | 0:02±0:00 | 0:15±0:00 |
| POSTERIOR SOURCE GP | 1:11±0:02 | 2:46±0:01 |
| FORWARD ODE INTEGRATION | 2:11±0:01 | 3:45±0:41 |
| POSTERIOR SAMPLING (TOTAL) | 3:24±0:03 | 6:46±0:02 |

*Table 4.* Computation time of the different methods. The results are reported in (hours : )minutes : seconds and we display the mean±std over 16 different runs. Every experiment is computed on a single NVIDIA L40s (48 GB memory).

## C. Additional results for the image regression tasks

In this section, we give the details of implementation for the different models in the image regression tasks, as well as an ablation of the different parameters. We develop the details for RP Flow as well as the RFF Network and the GP Regressors which can be considered as ablations of our method.

### C.1. Implementation details

This experiment is conducted on 32 images from the validation set of the Div2K dataset. Specifically, 16 of these images are sampled at random and used to tune the hyperparameters of each model, and the remaining 16 are used to report the results in the main paper. Each image is cropped at its center to a $1024 \times 1024$ patch that is resized to a $512 \times 512$ grid using bilinear resampling. For the $4\times$ upsampling task, the models are trained on a subsampled grid of pixels, and on $25\%$ randomly sampled pixels for the "Random" task. The metrics are computed on the remaining pixels, except the SSIM which is computed on the $512 \times 512$ reconstructed image. The visual samples from Figures 3, 13, 12 and 14 are shown in the $4\times$ upsampling task, on a subsampled grid with an offset of one pixel in each direction.

Both RP Flow and the RFF Network are parameterized by a 4 layers MLP with ReLU activations (sigmoid is used as an output function for the RFF MLP), and trained with the MSE loss for 10000 iterations using the Adam optimizer (Kingma & Ba, 2015) with default parameters and a learning rate of $10^{-3}$. We tried to match the experimental setup introduced by Tancik et al. (2020) as much as possible. However, RP Flow converges more slowly than the RFF Network, and we decided to train all models over more iterations. We made sure that these additional epochs did not introduce overfitting into

the baselines. To match the architecture' size between models, the RFF Network receives $512$ RFF frequencies while RP Flow receives $256$, along with an embedding of $t \in [0, 1]$, and $z_t(x)$ solution of the flow at time $t$. The GPR baseline is considered with a RBF kernel to match the Random Fourier Features and its lengthscale is tuned to maximize PSNR. Since we have access to additional images for tuning the models' hyperparameters, we choose to tune the models with respect to PSNR rather than MLE, although the latter could also be applied in a manner analogous to Gaussian Processes.

### C.2. Ablation study

The main hyperparameters for each model are its lengthscale ($\sigma_{RFF}$ for RP Flow and the RFF Network and $\sigma_{GP}$ for the GPs) and the calibration parameter $\sigma_{Noise}$ ($\sigma$ in the main paper). For each model, we adopt a two-steps strategy where we first tune the lengthscale to maximize PSNR/SSIM and fix it to the optimal choice to tune $\sigma_{Noise}$. Figure 8 displays PSNR and SSIM as functions of the lengthscale of the model, on the 16 training images. Each model exhibits a range of lengthscales that yield high reconstruction quality, but the optimal choice is different for each model. For RP Flow, as we will sample from the posterior to maximize reconstruction quality, we use $\sigma_{RFF} = 40$, an optimal value that is lower than when considering prior samples. For the RFF Network, we recover the optimal $\sigma_{RFF} = 64$ found in the original paper (they rescale $\sigma_{RFF}$ by a factor $2\pi$, which leads to an optimal value of 10 in their paper). Finally, the GP lengthscale is chosen in spatial domain instead of frequency, and we found $\sigma_{GP} \approx 0.007$ to be optimal. Figure 13 shows samples from each model across different lengthscales.

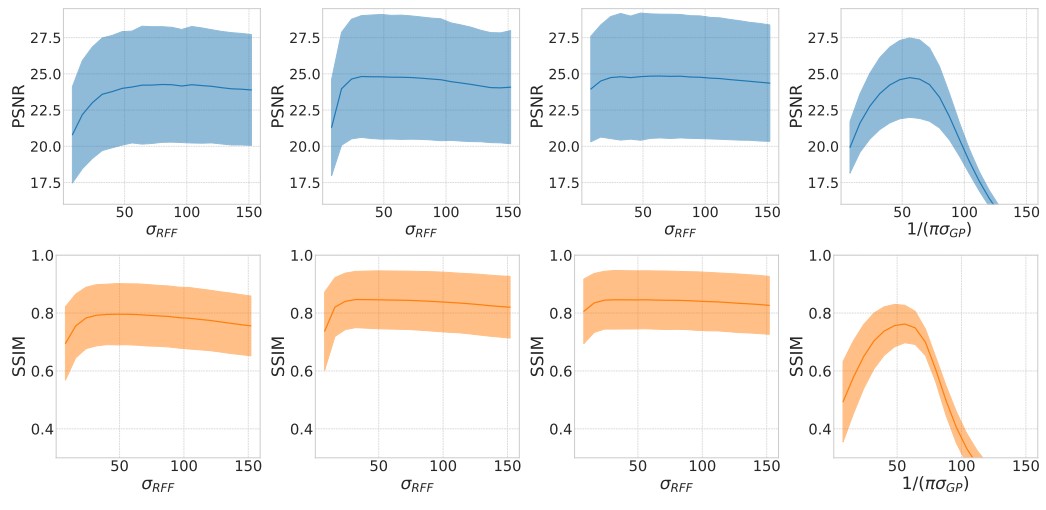

RP Flow prior samples  RP Flow posterior samples  RFF MLP regression  GP posterior samples

*Figure 8.* PSNR and SSIM as a function of model's lengthscale ($\sigma_{RFF}$ for RP Flow and RFF MLP ; $\sigma_{GP}$ for the GPR) on the 16 training images. Mean metric is represented as a solid line and standard deviation in light envelope.

Once RP Flow is tuned to maximize the reconstruction quality, we tune its regularization parameter $\sigma_{Noise}$ ($\sigma$ in the main paper) to achieve calibrated predictions in a similar fashion to Gaussian Process regression. We find that $\sigma_{Noise} = 0.06$ to be optimal for RP Flow as well as the GP baseline. Figure 9 displays the metrics as a function of $\sigma_{Noise}$. It is generally observed that, for all models, permitting noisy predictions leads to degraded reconstruction performance, resulting in lower PSNR and SSIM values. However, in the case of RP Flow, the posterior sampling strategy enables the recovery of high-quality samples, as evidenced by the stable PSNR and SSIM values shown in Figure 9. This effect is further illustrated on a training image in Figure 14.

To sample from RP Flow's posterior, we need to tune another parameter ($\sigma_{GP_{posterior}}$), that is used as the lengthscale of the source posterior GP. In Figure 10, we show the evolution of the metrics as a function of this parameter, for three different choices of $\sigma_{RFF}$. We found the optimal parameter to be $\sigma_{GP_{posterior}} = 0.008$, independently of the initial model's lengthscale. Interestingly, the choice of $\sigma_{GP_{posterior}}$ that maximizes PSNR is generally higher-frequency than the one induced by RFFs. This is likely due to the fact that real data usually don't follow exactly a Gaussian covariance and high-frequencies are not well understood during training. We find that even with a low-frequency initial lengthscale $\sigma_{RFF}$, it is possible to boost the posterior metrics with a higher-frequency source posterior.

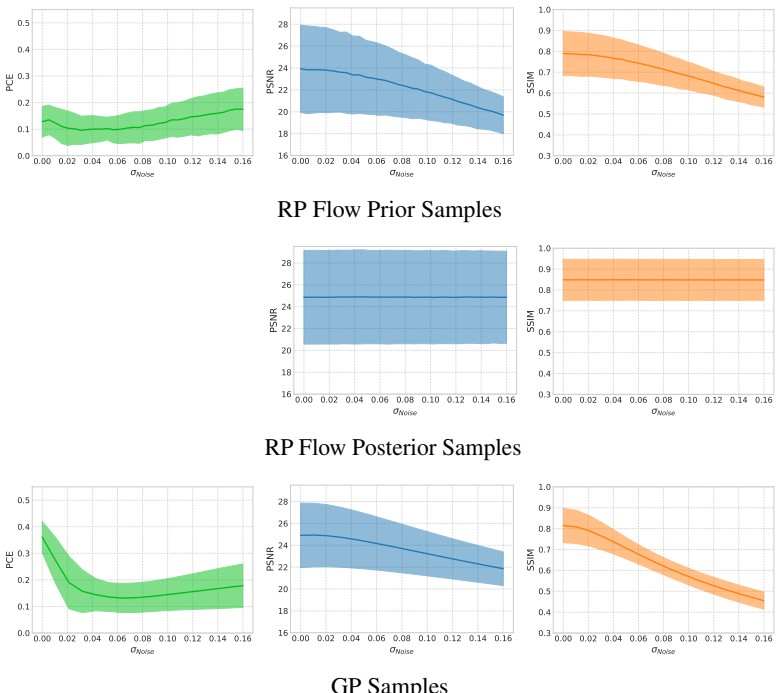

RP Flow Prior Samples

RP Flow Posterior Samples

GP Samples

*Figure 9.* Metrics as a function of $\sigma_{Noise}$ ($\sigma$ in the main paper) for RP Flow and GPR in the image upsampling task. Modeling noise in the observed data enables to calibrate the models but reduces the quality of the samples. Samples from RP Flow posterior stay consistent in quality when considering noisy observations, enabling the modeling of calibrated uncertainties without compromising in sample quality.

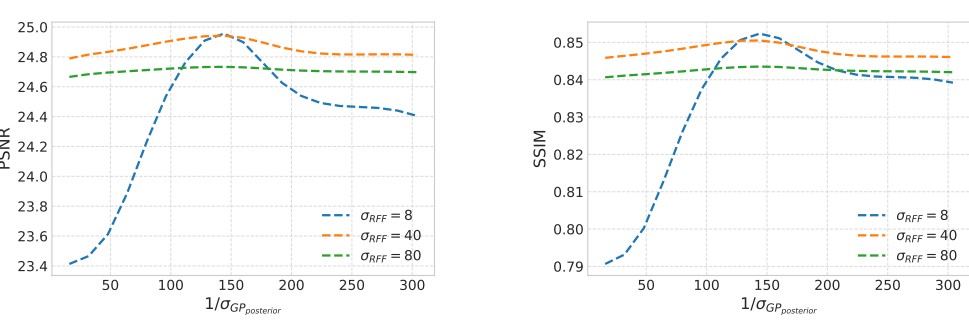

*(a)* PSNR as a function of noise correlation.

*(b)* SSIM as a function of noise correlation.

*Figure 10.* PSNR and SSIM of Posterior RP Flow as a function of Gaussian Process lengthscale, for different model's RFF embedding. The different models show a maximum metric around $\sigma_{GP_{posterior}} = 0.012$, independently of the model RFF embedding. High frequency in the noise tend to mitigate the spectral bias induced by a low frequency RFF embedding.

The choice of Random Fourier Features is motivated by the modeling of the source space by a Gaussian Process. However, in practical applications, there is no guarantee that this modeling choice yields the best predictive performance. In Table 5, we compare the version of RP Flow presented in the main paper with ablated variants in which the positional conditioning is either removed or replaced with a standard positional encoding. In both image regression settings, we observe, consistent to the INR case (Tancik et al., 2020), that RP Flow achieves its best performance when RFFs are used for positional conditioning.

Finally, we evaluate the performance of RP Flow on a compression task and compare it against a Bayesian INR. We report the results in Table 6. In this setting, RP Flow is parametrized with a RFF MLP, as in the main experiments of the paper, with the same choice of hyperparameters. The Bayesian INR is parametrized by a SIREN Network, following the configuration described in the original paper. All metrics are computed on the training pixels only, and samples are drawn from the prior of RP Flow. Interestingly, while the Bayesian INR achieves a higher PSNR, RP Flow attains better SSIM and PCE scores,

| | UPSAMPLING | | RANDOM | |
| --- | --- | --- | --- | --- |
| METHOD | PSNR↑ | SSIM↑ | PSNR↑ | SSIM↑ |
| RP FLOW (NO POSITION CONDITIONING) | 21.72±3.60 | 0.58±0.12 | 21.41±3.58 | 0.57±0.12 |
| RP FLOW (POSITIONAL ENCODING) | 24.12±4.47 | 0.70±0.10 | 23.51±4.35 | 0.68±0.10 |
| RP FLOW (RANDOM FOURIER FEATURES) | **25.30±5.05** | **0.82±0.08** | **23.70±4.64** | **0.76±0.09** |

*Table 5.* Ablation study of RP Flow's positional encoding. Reconstruction metrics for the image regression tasks are reported for different choices of positional conditioning.

| METHOD | PSNR↑ | SSIM↑ | PCE↓ |
| --- | --- | --- | --- |
| BAYESIAN INR | **28.62±8.37** | 0.79±0.32 | 0.38±0.08 |
| RP FLOW | 26.11±1.65 | **0.88±0.03** | **0.18±0.06** |

*Table 6.* Quantitative results of the compression task on the images dataset. Results are reported as mean ± standard deviation over 16 evaluation images. For RP Flow, samples are drawn from the prior.

indicating a stronger capture of image structure at the expense of increased pixel-wise error.

## D. Additional results for the seismic interpolation task

In this section, we present the implementation details of the seismic interpolation task, followed by the results obtained under various training configurations. We develop the details for RP Flow as well as the RFF Network and the GP Regressors which can be considered as ablations of our method.

### D.1. Implementation details

The seismic interpolation experiments are conducted on two different datasets. The first is a synthetic dataset comprising 32 seismic volumes, each with dimensions $513 \times 513 \times 256$. Each volume is resampled using bilinear interpolation, resulting in traces of dimension 128. Among these 32 volumes, 16 are utilized for model tuning, while the remaining volumes are reserved for reporting the final results. The second dataset corresponds to the real F3 seismic dataset (Silva et al., 2019). For evaluation purposes, a sub-volume of dimensions $513 \times 513 \times 128$ is extracted. Additionally, another sub-volume of dimensions $128 \times 128 \times 128$, located on the opposite side of the full seismic volume, is employed to tune the model hyperparameters.

The PSNR and PCE are computed on all traces unseen during training. The SSIM is computed on the whole volumes and the Wasserstein distance is computed between sets of 512 randomly sampled traces from the ground truth and predictions.

RP Flow is parametrized by 1D U-Nets modified from Dhariwal & Nichol (2021) with different sizes. Specifically, the model is a U-Net with 1D ResNet blocks (He et al., 2016), Attention blocks (Vaswani et al., 2017), and SiLU activations (Elfwing et al., 2018). Similarly to class-conditioned models, the RFF positional embedding is concatenated to the time dimension. The three instances of RP Flow (S, M, L) are built with increasing depth and width to match 1.7M, 10M and 50M parameters to demonstrate underfitting, well-tuned and overfitting models. We tuned each model on 16 volumes and find optimal parameters independently of the size to be $\sigma_{RFF} = 6, \sigma_{Noise} = 0.012, \sigma_{GP_{Posterior}} = 0.025$ in the synthetic case and $\sigma_{RFF} = 6., \sigma_{Noise} = 0.05, \sigma_{GP_{Posterior}} = 0.025$ for the real noisy volume. The RFF Network is parametrized by a RFF MLP with 10 layers and 1024 hidden dimension, ReLU activations and a Sigmoid output function. The model is dimensioned to match the number of parameters of the optimal RP Flow M. We find the optimal lengthscale parameter to be $\sigma_{RFF} = 10$ in both the synthetic and real cases. The GPs are used with a Gaussian kernel in a slice-by-slice manner, as it is typically done in geosciences to reduce computation costs and issues related to anisotropy. The same code is used for the GPR baselines and the creation of RP Flow posterior processes. For the baselines, we find optimal parameters to be $\sigma_{GP} = 0.025$ for the noiseless GP and $\sigma_{GP} = 0.25, \sigma_{Noise} = 1.0$ for the calibrated GP.

All Deep Learning models are trained for 10000 steps on batches of 1024 randomly sampled traces, using a MSE loss and the Adam optimizer (Kingma & Ba, 2015) with default parameters. A learning rate of $2 \times 10^{-4}$ is used for RP Flow and $1 \times 10^{-3}$ for the RFF MLP. A warm-up phase increases linearly the learning rate to its final value over the first 2500 steps. Exponential Moving Average (Polyak & Juditsky, 1992) is used with a decay of 0.999 for a more stable training. The

models training and evaluation are computed on a single NVIDIA L40s (48 GB memory) GPU although the models can also run on smaller GPUs. All the code is written using the PyTorch (Paszke et al., 2019) library.

We note that the Deep Gaussian Process baseline, implemented using the GPyTorch library (Gardner et al., 2018), relies on a sparse variational approximation with a fixed set of inducing points to ensure computational tractability. While this approximation is effective in many practical settings (e.g. Image Regression), it can introduce limitations when modeling highly stationary processes such as seismic volumes. In our experiments, we observed that, despite careful tuning of the number of layers, inducing points, and variational parameters, training often converged to poor representations for such data. We hypothesize that this behavior stems from the difficulty of capturing fine-grained, globally repetitive structure using a limited set of inducing locations, which may lead to an overly smooth or degenerate posterior approximation.

### D.2. Additional results

In the main paper, we chose to evaluate the models using a setting with a regular training grid composed of one line every 64 units in each direction. This choice was made to reflect the level of sparsity typically observed in offshore wind farm seismic acquisitions. However, in practice, 2D grids can vary significantly: they may be denser or sparser, irregular, or even non-orthogonal. In this section, we evaluate the robustness of RP Flow to different training settings.

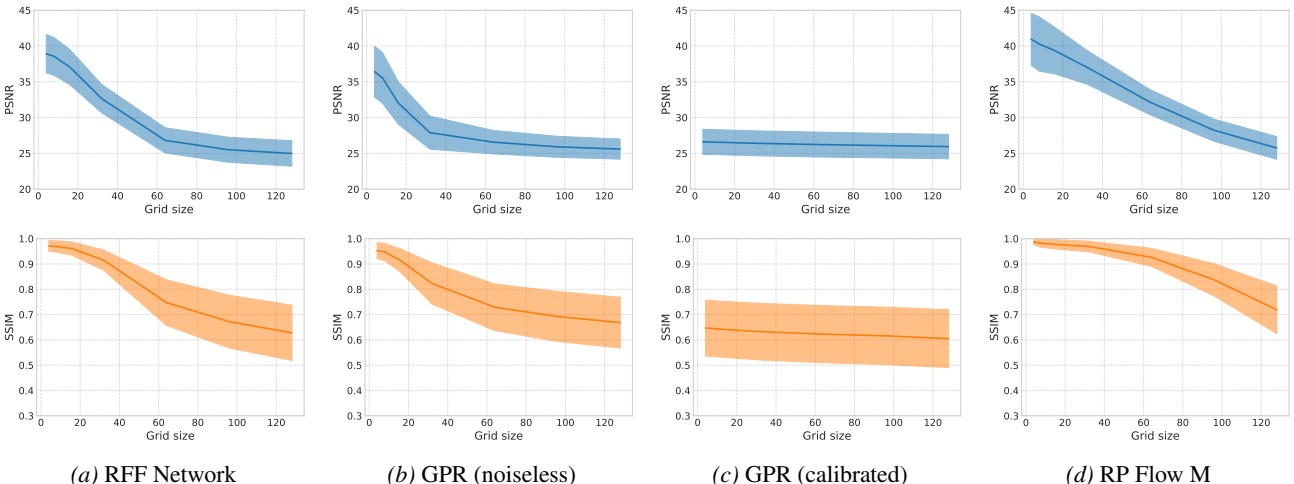

*(a)* RFF Network      *(b)* GPR (noiseless)      *(c)* GPR (calibrated)      *(d)* RP Flow M

*Figure 11.* Evolution of the reconstruction metrics as a function of the grid size $n$, in the case of a 2D training grid of one seismic line every $n$ in both directions.

In the first experiment, we maintain the regular grid assumption and investigate the effect of different line spacings. Figure 11 displays the reconstruction metrics as a function of the grid size for the different models. For dense grids, RP Flow is able to reconstruct high-quality volumes, while the reconstruction quality gradually degrades as sparsity increases. Nevertheless, RP Flow still performs reasonably well even in very sparse scenarios. We stops this experiment at a grid size of 128, as beyond this point the number of training samples becomes insufficient for efficient learning with Flow Matching, given a volume of size $513 \times 513 \times 128$. Notably, RP Flow consistently outperforms the considered baselines, even in dense-grid regimes where Kriging (noiseless GPR) is typically regarded as state of the art.

In the second experiment, we maintain the same ratio of training traces as in the setting presented in the main paper ($\sim 4 - 5\%$), but consider two alternative configurations for the training grid. In the first, denoted as *Random*, 5% of the traces are sampled at random for training. The second configuration, denoted as *Irregular*, uses a non-uniform grid where the spacing differs along the two axes: one line every 32 units in one direction and one line every 128 units in the other. Table 7 displays the results of the different models in these configurations.

The *Random* configuration provides better information to the models than the grid setups, as it is more likely to cover a greater diversity of traces during the training phase compared to regular grids, and the average distance between training and evaluation traces is smaller. This trend is evident in the results, where the RFF Network, GPR (noiseless) and RP Flow achieve high PSNR and SSIM metrics, whereas the calibrated GP perform similarly to the previous experiments. Although this configuration is not common in practice, it demonstrates that RP Flow does not rely on assumptions regarding the

| | RANDOM | | | IRREGULAR | | |
|---|---|---|---|---|---|---|
| METHOD | PSNR↑ | SSIM↑ | PCE↓ | PSNR↑ | SSIM↑ | PCE↓ |
| RFF NETWORK | **37.46**±**2.99** | 0.95±0.03 | | 28.68±2.21 | 0.83±0.08 | |
| GPR (NOISELESS) | 33.41±2.86 | 0.92±0.04 | 0.12±0.06 | 26.96±2.58 | 0.74±0.12 | 0.33±0.05 |
| GPR (CALIBRATED) | 26.41±1.70 | 0.63±0.11 | 0.11±0.06 | 26.28±1.34 | 0.62±0.11 | 0.12±0.06 |
| RP FLOW M ($\#params = 10M$) | 37.26±2.19 | **0.96**±**0.02** | **0.06**±**0.05** | **33.87**±**2.43** | **0.94**±**0.04** | **0.07**±**0.04** |

*Table 7.* Quantitative results of the seismic interpolation task on the synthetic dataset for two different training configurations. *Random* is the configuration where 5% of the total traces are sampled at random to train the models and *Irregular* is the configuration where the training traces are placed on an irregular training grid of one line every 32 in one direction and one line every 128 in the other. Results are reported as mean ± standard deviation over 16 evaluation volumes. In both cases, RP Flow achieves better performances than in the experimental setup presented in the main paper although the number of training traces stays approximately the same.

spatial regularity of training points. This capability can be beneficial for other tasks, such as handling corrupted traces in preprocessed seismic data (Mandelli et al., 2019). In the *Irregular* grid setup, the reconstruction results are slightly better than the regular one for the RFF Network and RP Flow but stay similar for the GPs. These results are expected, as this configuration is very similar to the regular one. Nevertheless, it is encouraging to observe that RP Flow is robust to this setup, which is more representative of practical scenarios where data acquisition often occurs on irregular grids.

Interestingly, without modifying the calibration parameter, RP Flow achieves the same calibration performance as in the regular grid experiment, demonstrating its ability to reliably model calibrated uncertainties.

# E. Additional Figures

In this section, we provide additional visual results for both the image regression and the seismic interpolation tasks.

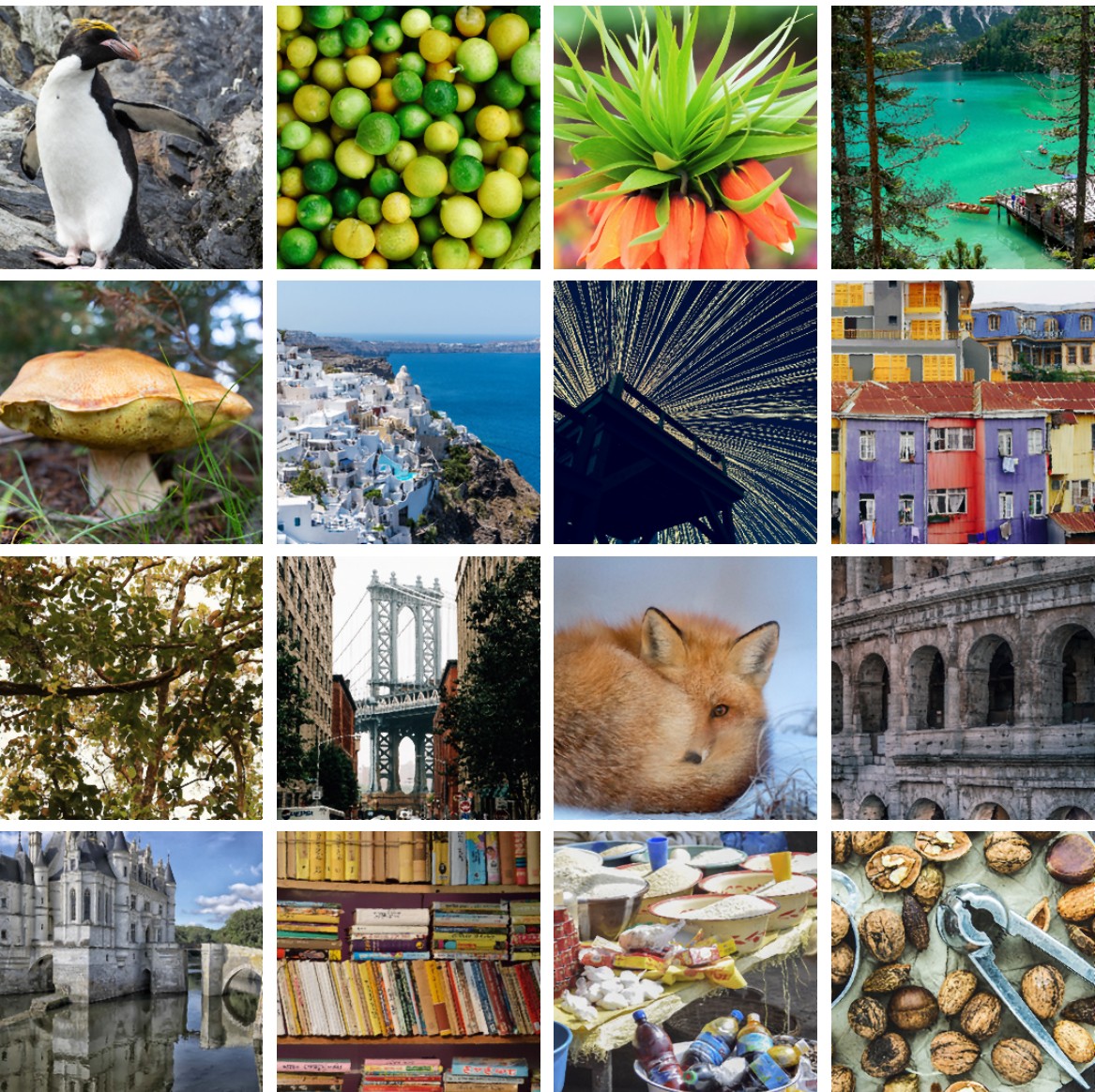

*Figure 12.* Samples from RP Flow's posterior on the 16 test images, in the $4\times$ upsampling task.

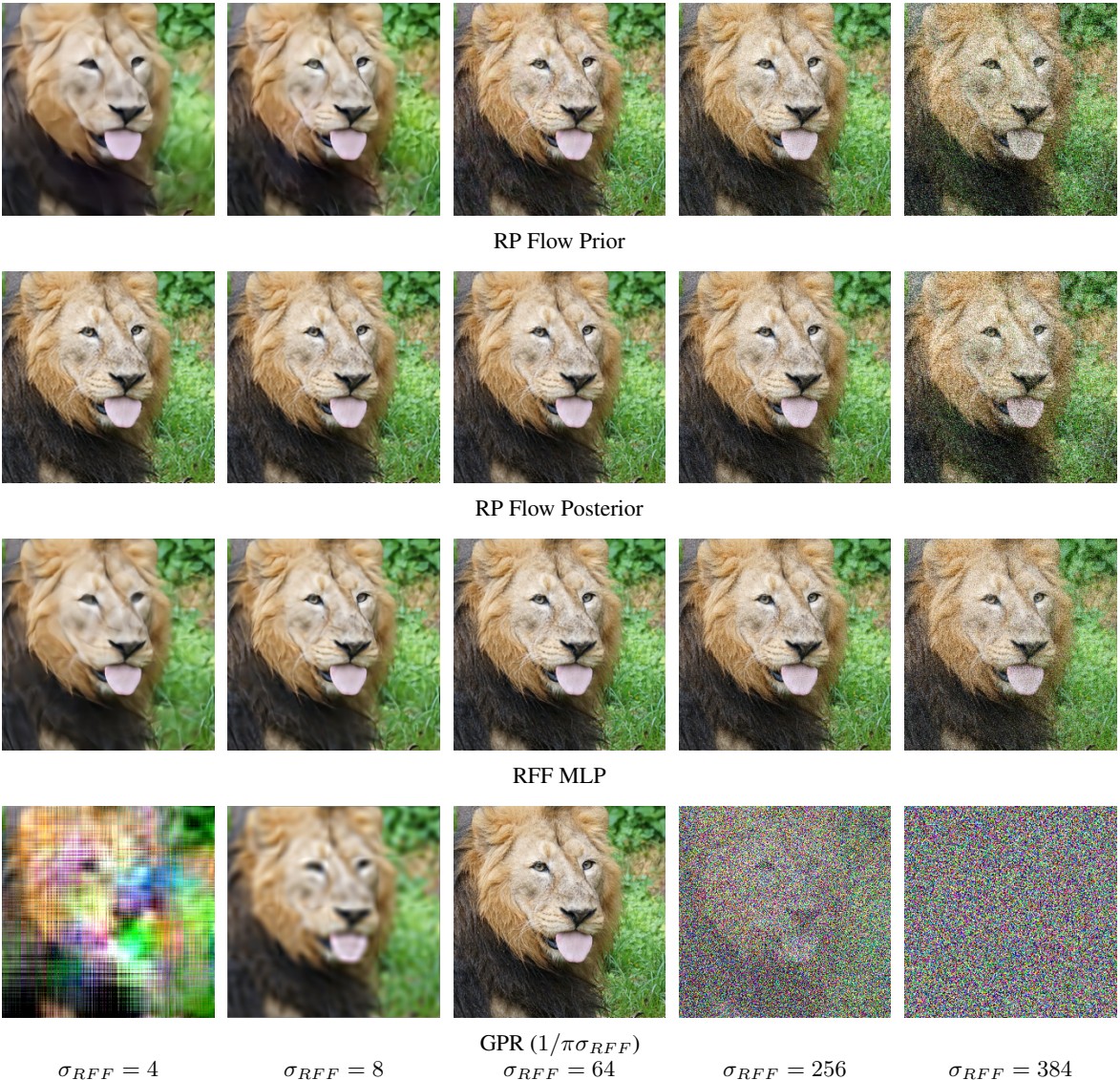

*Figure 13.* Samples from each model on a training image, for the $4\times$ upsampling task. Each column corresponds to a choice of model's lengthscale, in increasing order. For every model, low-frequency lengthscales predict smooth images and high-frequency ones introduce noise in predictions.

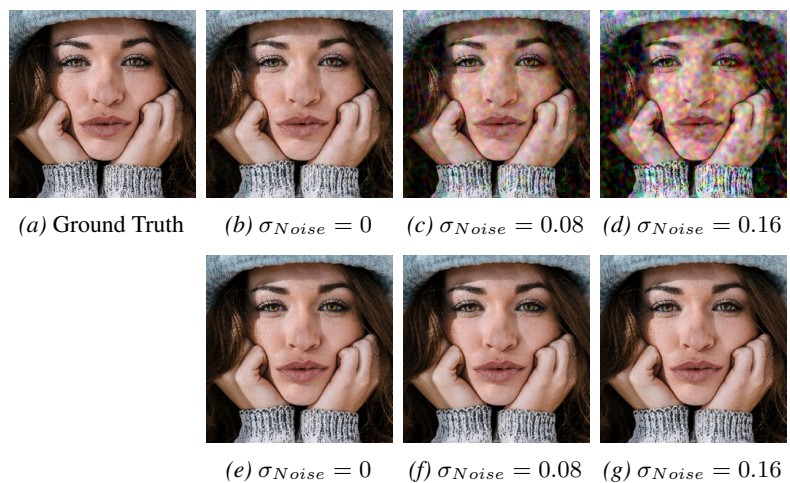

*(a)* Ground Truth   *(b)* $\sigma_{Noise} = 0$   *(c)* $\sigma_{Noise} = 0.08$   *(d)* $\sigma_{Noise} = 0.16$

*(e)* $\sigma_{Noise} = 0$   *(f)* $\sigma_{Noise} = 0.08$   *(g)* $\sigma_{Noise} = 0.16$

*Figure 14.* Samples from RP Flow prior (*(b, c, d)*) and posterior (*(e, f, g)*) on a validation image (*a*), in the image upsampling task, for different values of $\sigma_{Noise}$. Incorporating noisy observations during training degrades the quality of prior samples, while posterior samples remain unaffected.

*Figure 15.* Predictive distribution of randomly sampled vertical traces over the 16 test synthetic seismic volumes (top) and the real seismic volume (bottom) by RP Flow M. Ground truth trace is represented in dashed red, mean prediction in solid blue and standard deviation in light blue envelope.

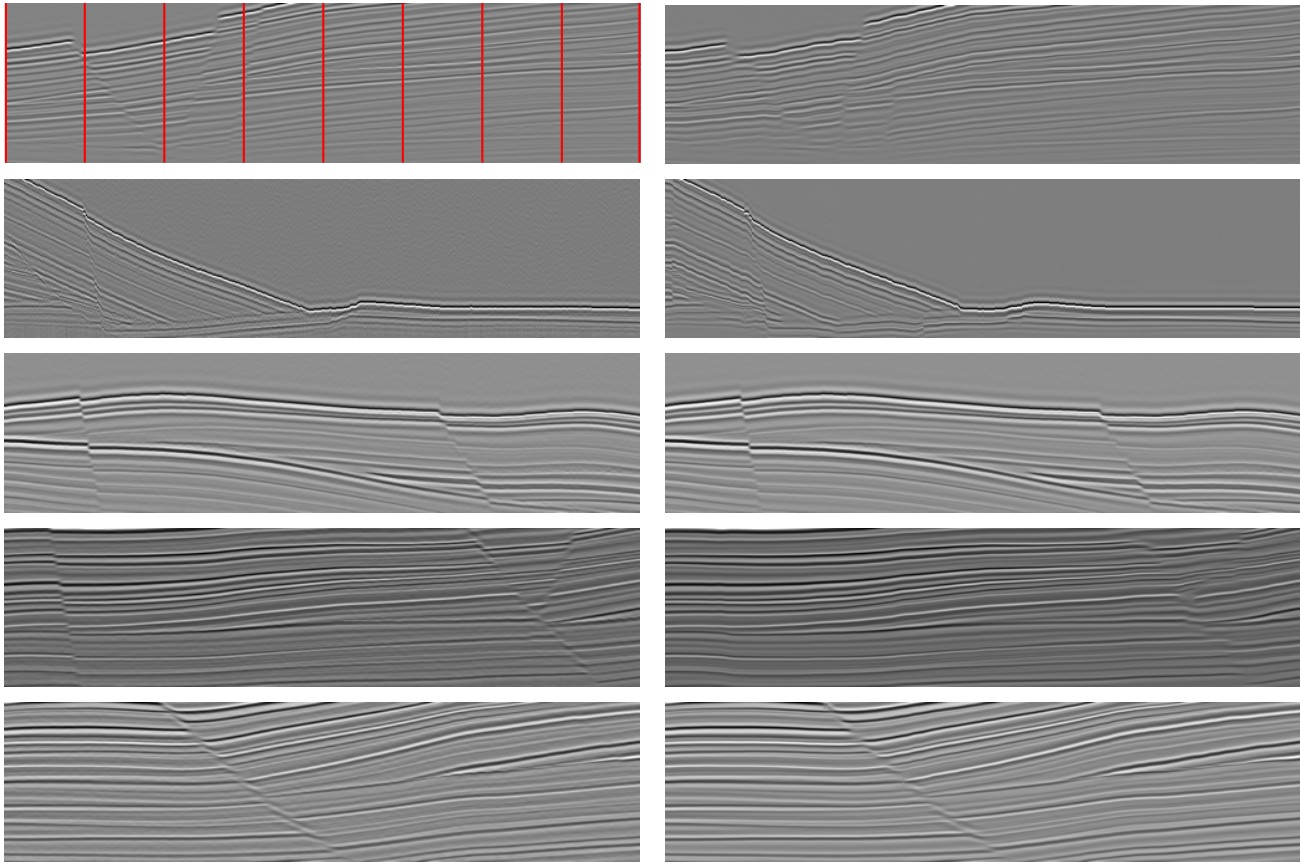

*Figure 16.* Randomly selected vertical sections from the 3D synthetic seismic test volumes. The ground truth section is shown on the left, while the predictions generated by RP Flow M are displayed on the right. Vertical traces that were seen during the model training are highlighted in red on the first GT section and is the same for the other sections.

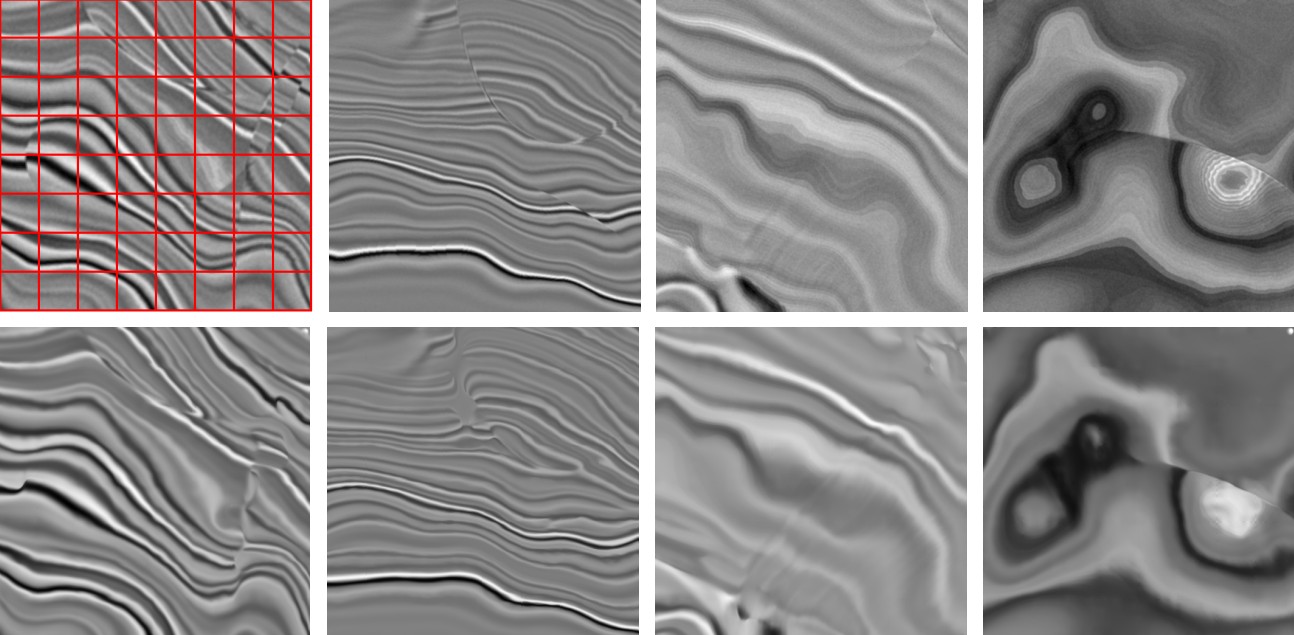

*Figure 17.* Randomly selected horizontal slices from the 3D synthetic seismic test volumes. The ground truth section is shown at the top, while the predictions generated by RP Flow M are displayed on the bottom. The training grid is highlighted in red on the first GT slice and is the same for the other slices.

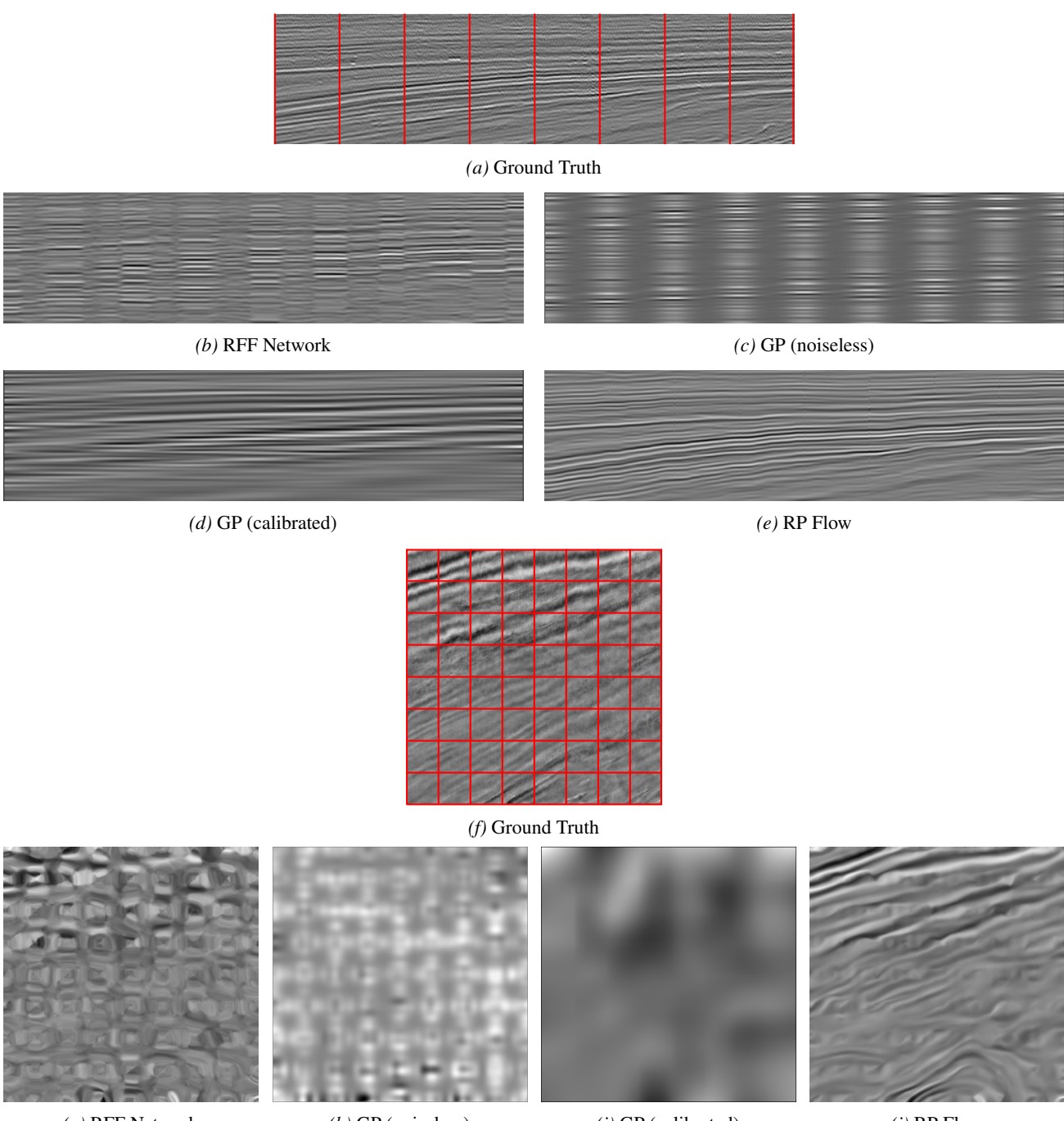

*(a)* Ground Truth

*(b)* RFF Network

*(c)* GP (noiseless)

*(d)* GP (calibrated)

*(e)* RP Flow

*(f)* Ground Truth

*(g)* RFF Network

*(h)* GP (noiseless)

*(i)* GP (calibrated)

*(j)* RP Flow

*Figure 18.* Vertical section (top) and horizontal slice (bottom) of the 3D real seismic volume. The voxels that were seen during the models training are highlighted in red on the Ground truth.

