# OpenReview forum: "Random Process Flow Matching: Generative Implicit Representations of Multivariate Random Fields"
_ICML.cc/2026/Conference — ICML 2026 regular_

### Official Review · Reviewer_JPn4 · 2026-03-03

**Soundness:** 3
**Presentation:** 3
**Significance:** 4
**Originality:** 3
**Overall Recommendation:** 4
**Confidence:** 3

**Summary:**

This paper proposes Random Process Flow Matching, a method designed for settings where only a single realization of a random field is available and the observations are sparse, irregular, and potentially noisy. The approach learns a generative model that can generate and interpolate multivariate random fields over a continuous spatial domain. It represents the target random field as the result of transporting a simple Gaussian source stochastic process through an invertible continuous flow defined by an ordinary differential equation trained with conditional flow matching. To capture spatial correlations and high frequency details in continuous coordinates, the method uses implicit neural representations together with random Fourier feature based coordinate encoding.

The main contributions are as follows. First, the paper introduces a stochastic process level flow matching framework that enables single sample, transductive generative field reconstruction from sparse observations. Second, it provides a practical posterior sampling strategy that maps observations back to the Gaussian source space by integrating the reverse ordinary differential equation, performs Gaussian process regression in the source space to obtain a conditional posterior and draw samples, and then maps those samples forward through the ordinary differential equation to the target space. This yields reconstructions that match the observations while producing meaningful uncertainty estimates. Third, the paper analyzes how regularity properties of sample paths transfer between the source and generated processes. Experiments on image super resolution, sparse pixel reconstruction, and seismic interpolation demonstrate strong performance in reconstruction quality and uncertainty calibration.

**Compliance With Llm Reviewing Policy:**

Affirmed.

**Key Questions For Authors:**

1. Your posterior sampling relies on integrating the reverse ordinary differential equation to map observations into the source space, performing Gaussian process regression in the source space, and then pushing samples forward through the ordinary differential equation. When using discrete time steps and in the presence of numerical solver error, to what extent can observation consistency and posterior unbiasedness still be guaranteed? Are there empirical or theoretical error bounds, or at least experimental results reporting how observation residuals and calibration metrics vary under different step sizes and solver configurations?

2. The choice of kernel and length scale in the source space Gaussian process regression may be critical for posterior uncertainty and interpolation quality. How are these hyperparameters selected in practice? Do you use data driven learning or marginal likelihood optimization, or are they fixed to heuristic values? Could you provide a more systematic sensitivity analysis and recommended default settings? If the hyperparameter selection is stable and does not require extensive tuning, it would substantially improve the method’s practicality and impact.

3. As the number of observed points increases, the output dimensionality becomes higher, or denser sampling is required, how does the total cost of reverse mapping plus Gaussian process regression plus forward sampling scale? Do you employ sparse approximations, inducing points, blockwise schemes, or local kernels to control complexity, and how do these approximations affect calibration and sample quality?

4. Random Fourier features and the source process design naturally align with stationary or approximately stationary structure. How does the method perform on clearly nonstationary random fields, strongly anisotropic fields, or fields with sharp discontinuities? Are extensions such as location dependent kernels, piecewise modeling, or mixtures of source processes necessary?

5. The method appears to require training or adaptation for each new field instance. Does training need to start from scratch, or can you transfer or share initialization across tasks from the same distribution, for example via meta learning, to reduce cost? In a multi instance, same distribution setting, how should RP Flow be positioned relative to conventional multi sample generative models?

**Limitations:**

yes

**Strengths And Weaknesses:**

# Soundness
* The overall method is self-consistent: the target random field is represented as the transport of a Gaussian source process via an invertible ODE flow, and the vector field is trained using conditional flow matching, with a clear technical pipeline.
* The posterior sampling procedure is well-motivated: it first maps inversely back to the source space, performs conditioning and sampling via Gaussian process regression, and then pushes forward to the target space. This approach strictly matches observations while quantifying uncertainty.
* The theoretical analysis of regularity transfer is meaningful but relies on assumptions such as Lipschitz continuity, which are difficult to verify in high-capacity neural vector fields. Moreover, the tightness of the bounds in practice is not sufficiently clear.
* Practical performance may depend heavily on the ODE solver and key hyperparameters. The paper could further strengthen its systematic characterization of numerical stability and robustness.

# Presentation
* The overall structure is clear. The distinction between prior sampling and posterior sampling is well explained, and explicit algorithmic procedures are provided.

# Significance
* The topic is practically important: sparse observations of single-field realizations are common in science and engineering, where the traditional multi-sample training assumption for generative models does not hold. This work has the potential to expand the applicability of generative models to such problems.
* The idea of conditioning in the source space is transferable and provides a general paradigm for combining invertible generative dynamics with classical Bayesian conditioning.

# Originality
* The main innovation lies in the organic integration and task-oriented restructuring of existing techniques, rather than the proposal of entirely new fundamental building blocks. Nonetheless, the compositional design is reasonable and well-expressed.
* Performing posterior conditioning in the source space is a distinctive design choice. Compared to heuristic uncertainty estimation only in the target space, it is more interpretable and systematic.

---

> ### Author Rebuttal · Authors · 2026-03-30
>
> We thank the reviewer for their careful reading and constructive feedbacks.
>
> Below, we respond to the main concerns raised by the reviewer.
>
> **On the Lipschitz assumptions.** Lipschitz continuity is a strong assumption, but in our setting $v_\theta$ is a neural network, for which it holds under standard conditions. Although the Lipschitz constant may be large, Theorem 3.3 only requires $v_\theta$ to be Lipschitz and does not depend on its magnitude. In contrast, the bounds in Theorem 3.4 are not expected to be tight when the Lipschitz constant is uncontrolled. Nevertheless, as discussed in Appendix B.3, the existence of an upper bound suffices for Monte Carlo estimation of moments, with tail bounds explicitly depending on this constant. Tighter bounds would require additional assumptions (e.g., bounded source or Gaussian/sub-Gaussian targets), which we intentionally avoid to remain general.
>
> **On the impact of the ODE Solver.** The choice of ODE solver and timestep count is important. Empirically, comparing Euler and Runge–Kutta solvers, we found that with aroung 50 timesteps all solvers are stable and numerical errors are negligible. In particular, backward–forward integration errors were typically smaller than those from posterior GP computation. Since the vector field is evaluated in parallel per pixel, inference is fast and increasing timesteps is not a limitation. Deriving general theoretical bounds on numerical instability seems difficult without additional assumptions, but we can include an empirical validation of unbiasedness across solvers and timesteps in the camera-ready.
>
> **On the tuning of hyperparameters.** The RFF lengthscale, GP kernel, and $\sigma_{noise}$ are indeed critical hyperparameters affecting interpolation quality and uncertainty calibration. As described in Appendix C.2, we tune them in a data-driven manner and find the model stable across a wide range of $\sigma_{RFF}$, while being more sensitive to $\sigma_{noise}$. When $\sigma_{RFF}$ is well tuned, performance is largely insensitive to the choice of GP posterior kernel (Fig. 11). Although this tuning can be heavy, we are exploring more systematic alternatives based on spectral analysis of high-resolution or subsampled data, which in recent experiments achieve comparable performance without extensive tuning. Following prior work (Tancik et al., 2020), we do not pursue optimizing Fourier features via gradient descent.
>
> **On the computational cost.** The reverse sampling step scales linearly with the number of observations but is performed only once. In practice, its cost is negligible compared to forward ODE integration, which dominates computation when generating multiple samples at evaluation locations.
>
> Computing the posterior GP does scale poorly with the number of observations and poses time and memory challenges. We evaluated inducing-point and blockwise approaches and found that blockwise schemes preserve quality as long as tile sizes exceed the kernel lengthscale. This is the strategy used in our experiments. In contrast, inducing points consistently degraded performance, both for posterior source process estimation and as a GP baseline in the target space. Finally, unlike GP regression directly in signal space, our method assumes channel-wise (or depth-wise) independence in multivariate settings, significantly reducing posterior GP costs without approximation. We refer to our response to Reviewer ycYh for further details.
>
> **On the extention to anisotropic/non stationary/discontinuous fields.** To remain in a general setting, we restrict this work to RFFs sampled from an isotropic Gaussian, which naturally aligns with stationary, isotropic fields. Nevertheless, similarly to INR Fourier Feature MLPs, RP Flow can represent anisotropic and nonstationary random fields in practice; natural images, which exhibit both properties, are modeled well empirically. RP Flow is fully compatible with extensions of RFFs, including anisotropic features, location-dependent kernels, and nonstationary Fourier features (JF Ton et al., 2018). In particular, first-order nonstationarity can be captured through model biases, and discontinuities can be handled without breaking stationarity assumptions, as shown by Theorem 3.3 and Appendix B.2.
>
> **On Meta-initialization.** The proposed method is transductive and therefore requires retraining for each new field instance. While RP Flow is compatible with meta-learning, our limited experiments with standard approaches such as MAML and Reptile did not reduce fine-tuning cost and, for first-order stationary fields (e.g., seismic volumes), even degraded reconstruction performance. We nonetheless view meta-learning extensions of RP Flow as a promising future direction. Regarding the second point, this work focuses on transductive settings; we refer the Reviewer to our response to Reviewer Ervj, which includes a comparison with large-scale inductive generative models for upsampling.

---

> > ### Author Rebuttal · Reviewer_JPn4 · 2026-04-01
> >
> > My concerns have been adequately addressed.

---

> > > ### Author Response · Authors · 2026-04-06
> > >
> > > We thank the reviewer for their response and are pleased that our previous reply helped clarify the main concerns.

---

### Official Review · Reviewer_ycYh · 2026-03-11

**Soundness:** 2
**Presentation:** 2
**Significance:** 2
**Originality:** 2
**Overall Recommendation:** 3
**Confidence:** 3

**Summary:**

The paper introduces a novel generative model framework, named Random Process Flow Matching (RP Flow), to model spatial processes and signals as realizations of random fields. In particular, it combines implicit neural representations (INR) with conditional flow matching (CFM) to learn a transductive, self-supervised model from a single, potentially sparse, realization of a target process. The authors propose the bi-directional mapping between a pre-defined Gaussian source model, parameterized via random Fourier features, and the target signal space. To obtain posterior samples at new locations, the method projects observed data back to the source space via reverse ODE iteration, and then performs Gaussian process regression in this source space. The authors evaluated the proposed method on image regression tasks and a 3D seismic data interpolation task.

**Compliance With Llm Reviewing Policy:**

Affirmed.

**Final Justification:**

While the authors have addressed my initial concerns regarding the baselines and modeling setup, I believe the paper still requires significant improvement in its presentation. I strongly believe that improving the paper's narrative and presentation to acknowledge and discuss the complexity limitations will strengthen the paper. Therefore, I would maintain my score.

**Key Questions For Authors:**

- Could you explicitly state the computational and memory complexity of the posterior sampling step (Algorithm 1)?
- Can you clarify the statement in Section 3.4 about "independence between variables"? Are you assuming a diagonal covariance matrix for the source GP? If so, does all the spatial structure come strictly from the learned transport map?

**Limitations:**

Yes

**Strengths And Weaknesses:**

**Strengths:**
- The paper is well-written and easy to follow.
- The idea of using a continuous normalizing flow (learned via CFM) to map a complex, non-Gaussian spatial target process into a tractable Gaussian source process for posterior inference is well-motivated.
- The ability to learn a generative model from a single incomplete realization without a need of a large dataset of fully observed signals is practical, especially for applications where obtaining multiple independent realizations of a system is difficult.
- The qualitative results, particularly in the seismic interpolation task clearly demonstrate that RP Flow preserves high-frequency spatial structures much better than standard GPR.

**Weaknesses:**
- The paper tackles the problem of spatial interpolation with uncertainty from sparse observations. This is highly related to the Neural Processes (NPs) and other modern methods tackling this problem.  However, the authors did not discuss and empirically compare to NPs. In the experiments, the authors mainly consider vanilla GPR and deterministic INR as baselines. Therefore, it is difficult to justify that the proposed method is actually competitive or state-of-the-art.
- A main motivation stated in the introduction is to overcome the cubic scalability issues of GPs. However, in Algorithm 1 and Section 3.4, the posterior is obtained by performing GPR on the projected source observations. In Section 3.4, the authors claim, "due to the independence between variables, it is possible to compute a posterior GP for each variable individually. This avoids the need for large matrix operations." This statement is highly confusing. If the variables are independent in the source space, how does the model capture long-range spatial correlations? If spatial correlation is captured by the RFFs, sampling from the posterior still requires operating on the covariance matrix of the observations. The exact computational complexity of this "source GPR" step is obfuscated.

---

> ### Author Rebuttal · Authors · 2026-03-30
>
> We thank the reviewer for their careful reading and constructive feedbacks.
>
> Below, we respond to the main concerns raised by the reviewer.
>
> **On the lack of modern baselines.** In the paper, we chose to compare RP Flow primarily to Fourier Features Networks and GP Regression as these are the closest methods to our approach. The lack of additional comparisons against recent advances on INRs and GPs can be misleading regarding the competitivity of our method. For additional results of our method againast more recent baselines, please refer to our response to Reviewer WGSb in which we compare RP Flow to Deep Gaussian Processes and Siren INRs. Regarding Neural Processes, these methods learn a prior over a class of processes and can then be described as meta-learning or inductive algorithms. RP Flow is however a transductive model that is retrained for each new field instance. The experimental focus of the paper was on comparisons with other transductive methods operating on a per-instance basis. That said, we agree that a broader empirical comparison with inductive models such as NPs would be valuable to further contextualize RP Flow. If possible, could the reviewer please share a publically available recent implementation of Neural Processes with which we could run the experiment before the end of the author/reviewer discussions.
>
> **On the independence between source variable.** We thank the reviewer for pointing out that the phrasing in Section 3.4 may be confusing, and we clarify it here. This work considers multivariate random processes, where dependencies between variables can be strong in the target space. For example, RGB values of a pixel drawn from an image are not independent. When performing GP regression directly in the signal space, this leads to issues regarding kernel design as this new dimension is usually highly anisotropic, and computational costs (developed in response 3.). However, in Algorithm 1, we perform GPR in the source space. The source process has by design marginals that are per-variable independent (channel-wise independent for images / depth-wise for seismic). Using this property, we propose to compute a posterior GP per variable instead of computing one posterior GP with channel-wise independent covariance. This leads to reduced computational costs as developed in the following paragraph, without the need for additional assumptions. Spatially, we still need to design a (non-diagonal) covariance matrix to compute the posterior source GP. The corresponding paragraph in Section 3.4 can be modified to improve quality in the camera-ready version.
>
> **On the complexity of Algorithm 1.** From a computational perspective, the main advantage of RP Flow over GP regression performed directly in the signal space lies in the per-variable independence in the source space, as discussed above. For a process with $n$ variables, RP Flow allows us to replace a joint computation with complexity in $O(n^3)$ to a $O(n)$ and decrease the memory complexity from $O(n^2)$ to $O(n)$.
>
> Below, we provide a detailed breakdown of the computational and memory complexity of Algorithm 1. We consider a 1D multivariate process with $n$ variables, evaluated at $N = N_{train}+N_{test}$ positions, and using k Euler steps to integrate the Flow Matching ODE:
>
> |       | Time Complexity | Memory complexity     |
> | :---:        |    :----:   |          :---: |
> | GP Regression      | $O((nN)^3)$       | $O((nN)^2)$   |
> | -        |    -   |  - |
> | Reverse ODE integration   | $O(kN)$        | $O(N)$      |
> | Posterior source GP   | $O(nN^3)$        | $O(nN^2)$ |
> | Forward ODE integration   | $O(kN)$        | $O(N)$ |
> | Total (Algorithm 1)   | $O((k+nN^2)N)$ | $O(nN^2)$ |
>
> In Section 3.4, we state: "*The source process
> has standard Gaussian marginals at each position. In the
> multivariate case, this property can significantly reduce the
> computational cost of computing the posterior source process compared to applying GPR directly in the target space.
> Specifically, due to the independence between variables, it
> is possible to compute a posterior GP for each variable individually.*"
> We emphasize that we do not claim to overcome the inherent cubic complexity of GP regression with respect to the number of spatial locations. Rather, the key benefit is that the posterior GP can be computed sequentially and independently for each variable, instead of jointly over all variables as would be required in the signal space. This leads to substantial computational and memory savings when the number of variables is large ; such as in the seismic interpolation experiments.

---

> > ### Author Rebuttal · Reviewer_ycYh · 2026-04-04
> >
> > The authors' rebuttal successfully addressed my primary empirical concerns by clarifying the transductive nature of their method and by providing necessary new baseline comparisons against Deep GPs and Siren. Furthermore, they resolved the contradiction in Section 3.4 by explicitly stating that the independence assumed in the source space applies across channels, rather than spatially. However, their provided complexity breakdown confirms my suspicion that the method still suffers from a strict $\mathcal{O}(N^3)$ spatial bottleneck, which contradicts the paper's introductory claims about overcoming GP scalability limits. I would raise my score, but it is strictly conditional on the final manuscript transparently acknowledging this cubic spatial scaling limitation.

---

> > > ### Author Response · Authors · 2026-04-06
> > >
> > > We thank the reviewer for their response and are pleased that our previous reply helped clarify the main concerns regarding the empirical results and independence across channels. Regarding complexity, we acknowledge that our method does retain the cubic complexity in the spatial dimension inherent to GP regression, and that we do not overcome this limitation. We agree that this point is important to communicate clearly. To address this, if accepted, we will revise the second paragraph of Section 3.4 to explicitly acknowledge the spatial cubic scaling as a limitation of our approach. In addition, we will include a detailed complexity analysis and empirical compute time measurements in an appendix (as referenced in our response to reviewer WGSb), to ensure full transparency. We would like to however emphasize again that our method still benefits from an independence across channels which reduces the memory footprint of the GP regression step, especially in settings where the number of channels is large (seismic data / multispectral imaging), as discussed in our first response. We appreciate the reviewer highlighting the importance of clearly stating this aspect, and we will make sure the final manuscript reflects this accurately.

---

### Official Review · Reviewer_Ervj · 2026-03-12

**Soundness:** 4
**Presentation:** 3
**Significance:** 3
**Originality:** 2
**Overall Recommendation:** 5
**Confidence:** 3

**Summary:**

The paper presents a methodology for posterior inference for imputing unobserved dimensions of time series. They allow for the modelling assumption that $Z(x) = T\# \Epsilon(x)$, where $T$ is a deterministic pointwise pushforward map $T(x, \omega) \mapsto T(x)$. This allows for easy sampling of an approximate posterior via integrating observed dimensions backwards in time before performing Gaussian process regression. They provide a cheap and simple approach which is relevant to tasks such as super-resolution.

**Compliance With Llm Reviewing Policy:**

Affirmed.

**Final Justification:**

The paper is sound, simple, and easy to follow. The limitations of the model are clear; however, it is well motivated within the problem settings the authors address. For this reason, I maintain my decision to accept.

**Key Questions For Authors:**

I would primarily like to see a comparison to standard diffusion model trained specifically trained for image super-resolution. While this might not be the most relevant comparison for the intended use case, it would be interesting to gauge relative performance of the method.

**Limitations:**

yes

**Strengths And Weaknesses:**

Pros:
The authors clearly state their methodology, provide ample justification to the algorithm, and present decent results on super resolution.
They present some relevant theory which elucidates the relation between the choice of prior and its impact on end generation.
The work presents itself clearly; and the problem statement is largely relevant to many fields.
The authors present convincing results against GPR regression.

Cons:
The model makes an assumption that there is a deterministic pushforward map, which makes the model good for super-resolution / random pixel inpainting; however, it loses universality of flow matching models.
They do not compare to diffusion + masked guidance which could practically be applied to their image benchmarks.
I found it unclear from the first read that the vector field acted deterministically pointwise, leading to some confusion over the algorithm. After another read this was more clear.

---

> ### Author Rebuttal · Authors · 2026-03-30
>
> We thank the reviewer for their careful reading and constructive feedbacks.
>
> Below, we respond to the main concerns raised by the reviewer.
>
> **On the deterministic pushforward map.** There are two regimes in which the model operates. During training, for a given position $x$, a sample is drawn from the source process marginal at position $x$ (chosen to be $N(0,I)$). To model noisy observations, gaussian noise with small variance $\sigma_{noise}^2$ is also added to the target observation $z_i$. This is done every epoch and independently in the source and target spaces. After training, the model does not yet act deterministically pointwise but transports a standard gaussian to a gaussian centered on the observations (at training positions, there are no guaranties of the exact model behavior outside the training observations).
> When inferring RP Flow with Algorithm 1, the model then behaves deterministically at training positions but remains stochastic outside.
>
> More generally, because RP Flow is designed to learn from a single realization of a random field, the observations at each position are inherently Dirac-distributed. This setting naturally restricts the universality one could expect from flow-matching models trained on large datasets of independent samples. An extension of RP Flow could be made in cases where multiple observations of the random process can be made. For example, if multiple trajectories of a dynamical system (such as repeated simulations of a pendulum with identical initial conditions) are observed, the model could be trained using multiple samples from the marginals while still performing inference on a single realization at test time.
>
> **On the comparison against diffusion + masked guidance.** In our different experiments, we focus on transductive baselines for our comparisons. Different supervised algorithms can however be used to perform image regression stochastically.
>
> Below, we show the results of a Flux.1 ControlNet model for masked guidance. Several issues can raise when using such models. This model is for example trained conditioned on bicubic-upsampled version of natural images, which leads to issues when evaluating it on *Random Pixel Inpainting*, thus we restrict the experiment to the 4X upsampling case. These models use guidance to condition on the low-resolution images but do not hardly constrain the generation to be correct on the low-resolution pixels. This leads to hallucinated samples that look visually appealing but are quantitatively far from the Ground Truth.
>
> |       | PSNR | SSIM     | PCE |
> | :---:        |    :----:   |          :---: | :---: |
> | Flux.1 ControlNet      | 17.93 $\pm$ 2.51      | 0.40 $\pm$ 0.10   | 0.07 $\pm$ 0.04 |
> | RP Flow | 25.30 $\pm$ 5.05       | 0.82 $\pm$ 0.08   | 0.09 $\pm$ 0.05 |
>
> The introduction of hallucinations leads to poor reconstruction metrics. However, the model seems to yield well calibrated uncertainty estimates.
>
> To look at the adaptability of such models, we tried to upsample (depth-by-depth) a seismic volume with the same model. Seismic data lie however far outside the distribution learned by the model and the results did not look visually coherent.

---

> > ### Author Rebuttal · Reviewer_Ervj · 2026-04-03
> >
> > I thank the authors for there rebuttal, my concerns have been addressed!

---

> > > ### Author Response · Authors · 2026-04-06
> > >
> > > We thank the reviewer for their response and are pleased that our previous reply helped clarify the main concerns.

---

### Official Review · Reviewer_WGSb · 2026-03-12

**Soundness:** 2
**Presentation:** 2
**Significance:** 2
**Originality:** 2
**Overall Recommendation:** 3
**Confidence:** 2

**Summary:**

The authors introduce Random Process Flow Matching (RP Flow) for regression of random fields, offering an alternative to Gaussian Process Regression (GPR). RP flow learns a transport map from one (potentially sparse) realization of the random field, to a simple Gaussian process prior, where inference takes place. The authors compare their approach to standard GPR and an implicit neural representation (INR) approach on an image task and a seismic imaging task from geology.

**Compliance With Llm Reviewing Policy:**

Affirmed.

**Final Justification:**

The authors have resolved my concerns regarding a fair empirical evaluation. However, having read the review of Reviewer ycYh together with the following discussion, I agree with the concern regarding the computational complexity of the proposed method, where the spatial complexity is not reduced compared with GP regression methods (as implied in the current version of the work). I have therefore decided to keep my original score.

**Key Questions For Authors:**

## Questions:

1. I have the impression that one could connect the work of the authors to deep GPs [Damianou et al., 2012] and its extensions (e.g. [Salimbeni et al., 2017]) with exactly one latent layer.
2. This work targets reconstruction under sparse data, but trains large models (as much as 50M parameters) on one sparse realization of as few thousand points. Still, the predictions of RP flow appear reasonably well-calibrated across all tasks. Could the authors provide intuition as to how RP flow avoids overfitting in these settings?
- Andreas Damianou, Neil D. Lawrence (2012). Deep Gaussian Processes. AISTATS
- Hugh Salimbeni, Vincent Dutordoir, James Hensman, Marc Deisenroth (2017). Deep Gaussian Processes with Importance-Weighted Variational Inference. ICML.

**Limitations:**

Limitations are not clearly discussed

**Strengths And Weaknesses:**

## Strengths:

1. The work is novel, and proposes an interesting approach to tackle the problem of regression on function spaces
2. The authors provide a clear way, justified by theoretical analysis, of adapting the prior over the source process to represent knowledge of the target process
3. The presentation is clear and easy to follow

## Weaknesses:

1. I believe that the empirical evaluation are a weaker aspect of this work. First, the baseline of GPR with calibrated noise is a valid comparison to run, but is also not representative of state of the art GP methods. A more fair comparison would be to also investigate GP regression methods that allow for more trainable parameters, e.g. deep kernels or deep GPs (see questions). Similarly, the INR baseline is very basic. While there are limited works on generative INR models, one could consider the works mentioned by the authors, or Bayesian INR models, as a stronger baseline (e.g. Guo et al. [2023])
2. The authors mention the computational limitations of GPR with high-dimensional data, but omit a discussion of the overall time for producing posterior samples. With RP flow, one first has to train a flow matching model before sampling, which I imagine is significantly slower for the experiments considered in this work.


- Zongyu Guo, Gergely Flamich, Jiajun He, Zhibo Chen, and José Miguel Hernández-Lobato. (2023). Compression with Bayesian implicit neural representations. NeurIPS.

---

> ### Author Rebuttal · Authors · 2026-03-30
>
> We thank the reviewer for their careful reading and constructive feedbacks.
>
> Below, we respond to the main concerns raised by the reviewer.
>
> **On the lack of modern baselines.** In the paper, we chose to compare RP Flow primarily to Fourier Features Networks and GP Regression as these are the closest methods to our approach. However, we agree that advances have been made both in GPs and INRs which would be valid baselines to assess the competitivity of our framework. Below, we report the results of a Deep Gaussian Process and a Siren INR for the image regression tasks.
>
> We consider:
> - A 2-Layer Deep GP with Gaussian Covariance, 12 hidden dimensions and 256 inducing points.
> - A Siren INR with 6 hidden layers and 256 hidden dimensions.
>
> |*4X Upsampling*|PSNR|SSIM|PCE|
> |:---:|:----:|:---:|:---:|
> |Deep GP|19.09 $\pm$ 2.93|0.36 $\pm$ 0.11|0.35 $\pm$ 0.04|
> | Siren|24.64 $\pm$ 4.87| 0.81 $\pm$ 0.09|
> |RP Flow|25.30 $\pm$ 5.05|0.82 $\pm$ 0.08|0.09 $\pm$ 0.05|
>
> |*Random*|PSNR|SSIM|PCE|
> |:---:|:----:|:---:|:---:|
> |Deep GP|19.08 $\pm$ 2.94|0.37 $\pm$ 0.11|0.36 $\pm$ 0.04|
> |Siren|22.19 $\pm$ 4.29|0.72 $\pm$ 0.11|
> |RP Flow|23.70 $\pm$ 4.64|0.76 $\pm$ 0.09|0.09 $\pm$ 0.04|
>
> Deep GP generates blurry predictions due its spectral bias and the high-frequency content in the images. Siren is able to reconstruct the high-frequencies in the images, yields high reconstruction metrics but lacks uncertainty. We point that RP Flow could also be parametrized by a Siren network but from our experiments, this leads to worse results.
>
> Bayesian INRs have seen limited use for image upsampling (e.g., Guo et al., 2023; He et al., 2024). Their posterior is defined only on the training coordinates, making them effective for compression but unsuitable for upsampling or reconstruction. We therefore compare a Bayesian INR to RP Flow only in the compression setting, where metrics are computed on the training positions. RP Flow is used as presented in the paper, with identical hyperparameters to the regression tasks; since the posterior inference algorithm is unavailable in compression, samples are drawn from its prior.
>
> |*Compression*|PSNR|SSIM|PCE|
> |:---:|:----:|:---:|:---:|
> |Bayesian INR|28.62 $\pm$ 8.37|0.79 $\pm$ 0.32|0.38 $\pm$ 0.08|
> |RP Flow|26.11 $\pm$ 1.65|0.88 $\pm$ 0.03|0.18 $\pm$ 0.06|
>
> RP Flow achieves lower PSNR but better structure (SSIM) and calibration (PCE). These results can be added to the appendix of our paper.
>
> **On connections with Deep GPs.** RP Flow and Deep GPs connect in their modeling choices. Both infer a distribution over functions to perform probabilistic inference in a transductive manner. However, they also differ in many ways that we describe as follows:
> - Each hidden state of a Deep GP is a Gaussian Process whereas RP Flow transports samples from a GP to a non-specified process which might not be gaussian.
> - The transport between source to target process is explicitely modeled in a Deep GP by the number of layers wheras RP Flow models this transport continuously by an ODE.
> - Access to posterior statistics is available from a Deep GP and RP Flow allows access only to samples from the target process.
>
> Overall, while both frameworks offer complementary strengths, we do not find straightforward connections between RP Flow and Deep GPs.
>
> **On the overall complexity.** We distinguish between two phases: training and sampling. Training is performed within the classical Flow Matching framework and does not require integrating the ODE. Moreover, the models operate in a position-wise manner, enabling the use of relatively small networks that work directly in signal space without relying on an autoencoder. From a memory standpoint, full images can be processed on most modern GPUs, and training can be efficiently batched to accommodate larger datasets, such as seismic volumes. In terms of runtime, training is relatively fast, requiring approximately 2 minutes per image and 5 minutes per seismic volume. Regarding sampling complexity, we refer to our response to Reviewer ycYh, in which we provide a detailed derivation for RP Flow.
>
> **On overfitting.** In the seismic interpolation experiment, we deliberately used a relatively large model (50M parameters) to demonstrate robustness to overfitting. We attribute this robustness to three main sources of regularization. First, the Flow Matching on its own heavily regularizes the training, as new gaussian noise samples are used every batches, forcing the model to explore its parameter space. Second, the choice of the RFF variance, $\sigma_{RFF}^2$, plays a crucial role in controlling model behavior: if the variance is too small, the model underfits the data, whereas if it is too large, generalization degrades. Finally, during training, signal observations are perturbed with additive Gaussian noise to model noisy measurements, which helps prevent the model from collapsing to a single deterministic solution.

---

> > ### Author Rebuttal · Reviewer_WGSb · 2026-04-03
> >
> > I thank the authors for addressing my concerns and answering my questions. I have a follow up question following the rebuttal from the authors. Regarding complexity: The authors provide a breakdown of the computational and memory complexities of sampling in response to reviewer ycYh, and here state that training costs are very small (2-5 minutes). The implication is that the cost of training is negligible compared to the sampling time. This sounds sensible to me, but I would still appreciate if the authors could explicitly provide the (approximate) wall-clock times for the different results reported in the main paper.

---

> > > ### Author Response · Authors · 2026-04-06
> > >
> > > We thank the reviewer for their response and are pleased that our previous reply helped clarify the main concerns. Below, we provide the measured wall-clock times for the different experiments.
> > >
> > > The setups are identical to those in the paper (Appendices C.1 and D.1). In particular, we report the times to draw the 64 samples used for PCE computation (on a single image/seismic volume), while factoring out every operation that needs to be performed only once (training, reverse ODE integration and computation of GP mean/covariance). We report the computation times in mean $\pm$ std over 16 different runs. In practice, the times needed to run the experiments heavily rely on the choice of a GPU and its ability to process large batches within memory. The results that we report are computed on a single NVIDIA L40s (48 GB memory). Results are reported in (hours **:** )minutes **:** seconds.
> > >
> > > ||Image upsampling|Seismic interpolation|
> > > |:---:|:---:|:---:|
> > > |Training|1:46 $\pm$ 0:02|5:53 $\pm$ 0:02|
> > > |Prior Sampling|2:11 $\pm$ 0:01|04:01:17 $\pm$ 0:49|
> > > |Reverse ODE integration|0:02 $\pm$ 0:00|0:15 $\pm$ 0:00|
> > > |Posterior Source GP|1:11 $\pm$ 0:02|2:46 $\pm$ 0:01|
> > > |Forward ODE integration|2:11 $\pm$ 0:01|03:59:37 $\pm$ 0:47|
> > > |Posterior Sampling (total)|3:24 $\pm$ 0:03|04:02:38 $\pm$ 0:47|
> > >
> > > Due to the limited time given for the author/reviewer discussion phase, we only consider one version (small) of RP Flow for seismic interpolation. If accepted, we will include these results along with those of the two other instances of RP Flow for seismic interpolation in the appendix.

---

### Decision · Program_Chairs · 2026-04-30

**Decision:**

Accept (regular)

**Comment:**

While reviewers agreed that the paper is well-written and sound, they also raised concerns about the paper's clarity/significance: reviewers agree on the lack of clarity regarding the computational complexity of the proposed method, where the spatial complexity is not reduced compared with GP regression methods (as implied in the current version of the work); lack of comparisons against several baselines models including deep GPs and neural processes. Some of these baselines were included in the rebuttal, but a more extensive comparison is needed.